# *Drosophila insulin-like peptide 2* mediates dietary regulation of sleep intensity

**Elizabeth B. Brown**[1], **Kreesha D. Shah**[1,2], **Richard Faville**[3], **Benjamin Kottler**[3], **Alex C. Keene**[1]*

**1** Department of Biological Sciences, Florida Atlantic University, Jupiter, Florida, United States of America, **2** Wilkes Honors College, Florida Atlantic University, Jupiter, Florida, United States of America, **3** Burczyk/Faville/Kottler LTD, London, England

* KeeneA@FAU.edu

**Data Availability Statement:** All relevant data are within the manuscript and its Supporting Information files. Original statistics and raw data will be published as a supporting file upon acceptance.

## Abstract

Sleep is a nearly universal behavior that is regulated by diverse environmental stimuli and physiological states. A defining feature of sleep is a homeostatic rebound following deprivation, where animals compensate for lost sleep by increasing sleep duration and/or sleep depth. The fruit fly, *Drosophila melanogaster*, exhibits robust recovery sleep following deprivation and represents a powerful model to study neural circuits regulating sleep homeostasis. Numerous neuronal populations have been identified in modulating sleep homeostasis as well as depth, raising the possibility that the duration and quality of recovery sleep is dependent on the environmental or physiological processes that induce sleep deprivation. Here, we find that unlike most pharmacological and environmental manipulations commonly used to restrict sleep, starvation potently induces sleep loss without a subsequent rebound in sleep duration or depth. Both starvation and a sucrose-only diet result in increased sleep depth, suggesting that dietary protein is essential for normal sleep depth and homeostasis. Finally, we find that *Drosophila insulin like peptide 2* (*Dilp2*) is acutely required for starvation-induced changes in sleep depth without regulating the duration of sleep. Flies lacking *Dilp2* exhibit a compensatory sleep rebound following starvation-induced sleep deprivation, suggesting *Dilp2* promotes resiliency to sleep loss. Together, these findings reveal innate resilience to starvation-induced sleep loss and identify distinct mechanisms that underlie starvation-induced changes in sleep duration and depth.

## Author summary

Sleep is nearly universal throughout the animal kingdom and homeostatic regulation represents a defining feature of sleep, where animals compensate for lost sleep by increasing sleep over subsequent time periods. Despite the robustness of this feature, the neural mechanisms regulating recovery from different types of sleep deprivation are not fully understood. Fruit flies provide a powerful model for investigating the genetic regulation of sleep, and like mammals, display robust recovery sleep following deprivation. Here, we find that unlike most stimuli that suppress sleep, sleep deprivation by starvation does not require a homeostatic rebound. These findings are likely due to flies engaging in deeper

**Funding:** This work was supported by NIH Grants 1R01 NS081512, 1R01 DC017390, and 1R01 HL143790 to ACK. The funders had no role in study design, data collection and analysis, decision to publish, or preparation of the manuscript.

**Competing interests:** I have read the journal's policy and the authors of this manuscript have the following competing interests: RF and BK are owners of BFK Labs that design and sell the Arousal Systems used in this paper. Both are also academic scientists and the analysis here required custom programming and analysis. Therefore, we feel that they meet the requirements outlined in the journal for authorship.

sleep during the period of partial sleep deprivation, suggesting a natural resilience to starvation-induced sleep loss. This unique resilience to starvation-induced sleep loss is dependent on *Drosophila insulin-like peptide 2*, revealing a critical role for insulin signaling in regulating interactions between diet and sleep homeostasis.

## Introduction

Sleep is a near universal behavior that is modulated in accordance with physiological state and environmental stimuli [1–3]. Numerous environmental factors can alter sleep, including daily changes in light and temperature, stress, social interactions, and nutrient availability [4,5]. A central factor defining sleep is that a homeostat detects sleep loss and then compensates by increasing sleep duration and/or depth during recovery [6,7]. However, recent findings suggest that the need for recovery sleep varies depending on the neural circuits driving sleep loss [8,9]. An understanding of how different forms of sleep restriction impact sleep quality and homeostasis may offer insights into the integration of an organism's environment with sleep drive.

Food restriction represents an ecologically-relevant perturbation that impacts sleep. In animals ranging from flies to mammals, sleep is disrupted during times of food restriction, presumably to allow for increased time to forage [10,11]. While the mechanisms underlying sleep-metabolism interactions are unclear, numerous genes have been identified that integrate these processes [12,13]. Further, neurons involved in sleep or wakefulness have been identified in flies and mammals that are glucose sensitive, raising the possibility that cell-autonomous nutrient sensing is critical to sleep regulation [14–16]. Despite these highly conserved interactions between sleep and metabolic regulation, little is known about the effects of starvation on sleep quality and homeostatically regulated recovery sleep when food is restored.

Fruit flies are a powerful model to study the molecular basis of sleep regulation [17,18]. Recently, significant progress has been made in identifying neural and cellular processes associated with detecting sleep debt and inducing recovery sleep [8,9,19–21]. However, a central question is how different genetic, pharmacological, and environmental manipulations impact sleep quality and homeostasis. Flies potently suppress their sleep when starved, and although the duration of starvation varies among studies, at least some evidence suggests they are resilient to this form of sleep loss [11,22,23]. Therefore, it is possible that starvation-induced sleep restriction is governed by neural processes that are distinct from other manipulations that restrict sleep.

Sleep quality is dependent on both sleep duration and sleep depth. The recent development of standardized methodology to measure sleep depth in *Drosophila* allows for investigating the physiological effects of sleep loss [24–26]. Mechanically depriving flies of sleep results in increased sleep duration and depth the following day, but the effects of other deprivation methods on sleep depth is largely unknown [24,26]. Applying these new approaches to quantify sleep during and following periods of starvation has potential to identify the mechanistic differences underlying resiliency to starvation-induced sleep loss.

Here, we find that starvation impairs sleep without inducing a homeostatic rebound, and that this is likely due to increased sleep depth during the period of food restriction. This phenotype can also be induced by feeding flies a diet lacking in amino acids, suggesting a role for dietary protein in maintaining normal sleep quality. Further, this resilience to starvation-induced sleep loss is dependent on *Drosophila insulin-like peptide 2*. Overall, these findings highlight the role of insulin signaling in regulating the interaction between diet and sleep depth.

## Results

To determine how different forms of sleep deprivation impact sleep depth and homeostatic recovery, we measured sleep and arousal threshold by video-tracking in the *Drosophila* ARousal Tracking (DART) system (Fig 1A; [25]). This system probes sleep depth by quantifying the responsiveness of sleeping flies to increasing intensities of mechanical stimuli (Fig 1A, representative stimulus train displayed on computer screen). After 24 hours of baseline sleep measurements in undisturbed female control ($w^{1118}$) flies, we restricted flies of sleep for 24 hours by caffeine feeding, administration of mechanical stimuli, or through starvation. The sleep- restricting stimulus was then removed at ZT0 the following day to measure rebound sleep duration and arousal threshold during recovery (Fig 1B). As expected, sleep restricting flies through the feeding of caffeine-laced food or mechanical stimulation resulted in a homeostatic rebound where sleep was increased during the 12 hr period following sleep deprivation (Fig 1C–1E). The timing of recovery sleep differed between the two deprivation protocols, with mechanical sleep disruption resulting in significant recovery as early as the first three hours following deprivation, while recovery sleep was only detected over the 12 hr period following caffeine feeding (Fig 1C and 1E). Both caffeine and mechanical sleep deprivation also resulted in increased arousal threshold during rebound, indicating deeper sleep (Fig 1D and 1F, right panels). To ensure that this increase in arousal threshold during recovery was not a result of habituation to the mechanical stimulus over the duration of the experiment, we measured arousal threshold on standard food over a 3-day period. We found no effect of time on sleep duration or arousal threshold (S1 Fig), suggesting that this increase in arousal threshold during rebound results from sleep restriction rather than habituation to the mechanical stimulus. Conversely, starved flies did not exhibit rebound sleep or increased arousal threshold during recovery (Fig 1G and 1H), although there is a trend towards a rebound in sleep duration during the first 3 hours of recovery. Together, these findings suggest starvation restricts sleep without inducing a homeostatic increase in sleep duration or depth during rebound.

To examine the effects of starvation-induced sleep restriction on sleep quality, we probed arousal threshold throughout the 24-hour starvation period (Fig 2A). Starvation significantly reduced sleep during both the day and night compared to flies maintained on standard fly food (Fig 2B–2C). Daytime arousal threshold did not differ between fed and starved flies; yet nighttime arousal threshold was significantly increased in starved flies (Fig 2D–2E). Quantifying reactivity to the mechanical stimulus as a function of individual sleep bout length revealed that arousal threshold was significantly higher in starved flies shortly after they enter sleep, suggesting starved flies are quicker to enter deeper sleep (Fig 2F). In addition, the increase in nighttime arousal threshold is specific to starvation, as there was no increase in nighttime arousal threshold when flies were sleep deprived using caffeine feeding or mechanical vibration (2G-J). Together, these findings suggest flies compensate for starvation-induced sleep loss by increasing sleep depth during the night.

To assess how sleep architecture is affected by these methods of sleep deprivation, we measured sleep using *Drosophila* activity monitors (DAM; [27]). Similar to our results obtained in the DART, we found that sleep is significantly reduced when flies are starved, mechanically shaken, and when fed caffeine. When flies are fed caffeine, the number of bouts remain unchanged, but their length significantly decreases (S2A–S2E Fig). When flies are mechanically sleep deprived, both bout number and length are significantly reduced (S2F–S2H Fig). Lastly, starvation-induced sleep suppression results from a significant decrease in bout number, but not bout length (S2I–S2K Fig). Given that deeper sleep states are correlated with longer sleep bouts [24,25,28], these findings provide additional support for our findings that sleep depth increases during starvation.

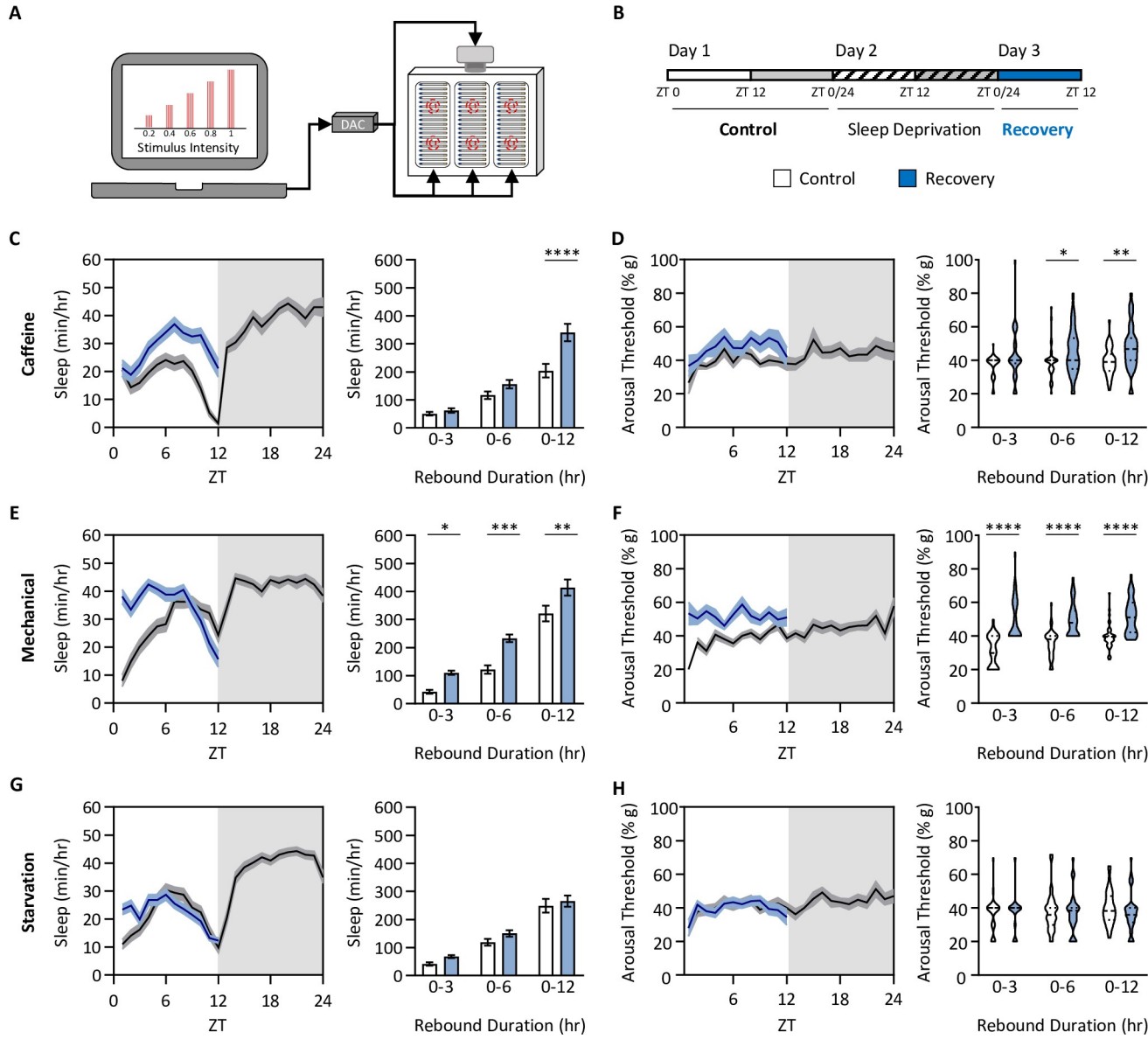

**Fig 1. Homeostatic rebound following sleep deprivation is treatment dependent.** (A) The *Drosophila* Arousal Tracking (DART) system records fly movement while simultaneously controlling mechanical stimuli via a digital analog converter (DAC). Here, mechanical stimuli are delivered to three platforms, each housing twenty flies. Mechanical stimuli of increasing strength were used to assess arousal threshold (shown on the computer screen). Arousal thresholds were determined hourly, starting at ZT0 [25]. (B) Total sleep and arousal threshold were assessed for 24 hrs on standard food (Control). Flies were then sleep deprived (Sleep Dep) for 24hrs using one of three treatments: 0.5mg/mL caffeine, mechanical vibration, or starvation, after which homeostatic rebound (Recovery) was assessed in the subsequent 12 hrs. Comparisons were made between the first 12 hrs of the control day (white box) and the homeostatic rebound (blue box). (C-H) Daytime sleep and arousal thresholds prior to and after 24 hrs of sleep deprivation. Profiles of sleep and arousal threshold are shown prior to the quantification of each trait. Measurements of homeostatic rebound were assessed in 3-, 6-, and 12-hr increments. (C,D) Sleep (C) and arousal threshold (D) significantly increases after 24hrs on standard food media containing 0.5mg/mL caffeine (Sleep: two-way ANOVA: $F_{1,228} = 16.76$, $P<0.0001$; Arousal threshold: REML: $F_{1,76} = 5.691$, $P = 0.0205$; N = 38–40). (E,F) Sleep (E) and arousal threshold (F) significantly increases after 24hrs of randomized mechanical vibration (Sleep: two-way ANOVA: $F_{1,234} = 35.11$, $P<0.0001$; Arousal threshold: REML: $F_{1,78} = 63.07$, $P<0.0001$; N = 40). (G,H) There is no difference in sleep (G) or arousal threshold (H) after 24hrs of starvation (Sleep: two-way ANOVA: $F_{1,210} = 3.272$, $P = 0.0800$; Arousal threshold: REML: $F_{1,70} = 0.0251$, $P = 0.8746$; N = 36). For profiles, shaded regions indicate +/- standard error from the mean. White background indicates daytime, while gray background indicates nighttime. For sleep measurements, error bars represent +/- standard error from the mean. For arousal threshold measurements, the median (dashed line) as well as 25th and 75th percentiles (dotted lines) are shown. * = $P<0.05$; ** = $P<0.01$; *** = $P<0.001$; **** = $P<0.0001$.

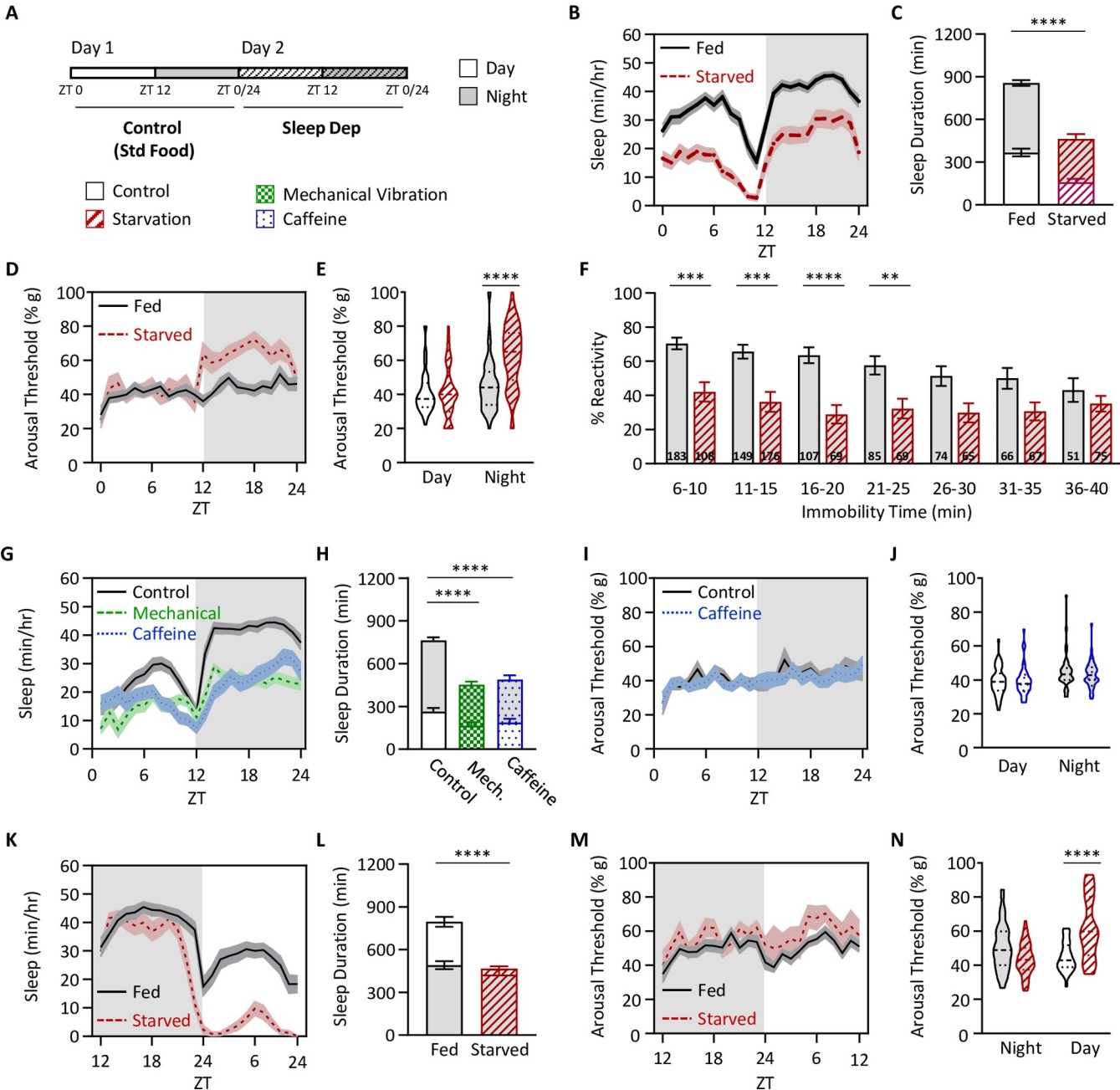

**Fig 2. Starvation increases arousal threshold.** (A) Total sleep and arousal threshold were assessed for 24 hrs on standard food (Control) and then starved for 24 hrs on agar (Sleep Dep). Flies were flipped to agar at ZT0. (B) Sleep profiles of fed and starved flies. (C) Sleep duration decreases in the starved state (two-way ANOVA: $F_{1,220} = 54.63$, $P<0.0001$), and occurs in both the day ($P<0.0001$) and night ($P<0.0001$). (D) Profile of arousal threshold of fed and starved flies. (E) Arousal threshold significantly increases in the starved state (REML: $F_{1,110} = 34.68$, $P<0.0001$), and occurs only at night (day: $P = 0.2210$; night: $P<0.0001$). (F) Sleep depth is not correlated with sleep duration in starved flies. The proportion of flies that reacted to a mechanical stimulus for each bin of immobility was assessed. Starved flies are significantly less likely to respond to a mechanical stimulus during sleep (ANOVA: $F_{1,1230} = 68.47$, $P<0.0001$), specifically when flies have been sleeping for less than 30 min (6–10 min: $P<0.0001$; 11–15 min: $P = 0.0002$; 16–20 min: $P<0.0001$; 21–25 min: $P = 0.0094$; 26–30 min: $P = 0.0538$). Numbers within each bar represent the frequency of individuals for each bin of immobility. All measurements were taken at night. (G-J) Sleep and arousal threshold measurements were taken over a 24 hr period from flies on standard food media or 1% agar. Flies were flipped to agar at ZT12. (G) Sleep profiles of fed and starved flies. (H) Sleep duration decreases in the starved state (two-way ANOVA: $F_{1,172} = 32.22$, $P<0.0001$), but occurs only during the day (day: $P<0.0001$; night: $P = 0.2036$). (I) Profile of arousal threshold of fed and starved flies. (J) Arousal threshold significantly increases in the starved state (REML: $F_{1,74} = 9.248$, $P<0.0027$) and occurs only during the day (day: $P<0.0001$; night: $P = 0.0960$). (K-N) Unlike starvation, other methods of sleep deprivation do not increase nighttime arousal threshold. Flies were sleep deprived for 24 hrs using either mechanical vibration or 0.5mg/mL caffeine. (K) Sleep profile. (L) Relative to control, sleep duration significantly decreases for each method of sleep deprivation (two-way ANOVA: $F_{1,234} = 21.58$, $P<0.0001$), and occurs only at night ($P<0.0001$). (M) Profile of arousal threshold. (N) Arousal threshold remains unchanged when using each of the described methods of sleep deprivation

(REML: $F_{1,72}$ = 0.2563, $P$ = 0.6142, N = 38). Arousal threshold via mechanical vibration was unable to be calculated since the mechanical vibration used to assess arousal threshold was being used to implement sleep deprivation. For profiles, shaded regions indicate +/- standard error from the mean. White background indicates daytime, while gray background indicates nighttime. For sleep measurements, error bars represent +/- standard error from the mean. For arousal threshold measurements, the median (dashed line) as well as 25th and 75th percentiles (dotted lines) are shown. ** = $P$<0.01; *** = $P$<0.001; **** = $P$<0.0001.

Sleep duration and resistance to starvation are sexually dimorphic and vary based on genetic background [29–32]. To determine whether the starvation-dependent changes in sleep depth are generalizable across sexes, we measured arousal threshold in starved male control ($w^{1118}$) flies. We found that starvation results in reduced sleep duration and increased night-time arousal threshold, phenocopying our results in females (S3A–S3D Fig). To determine if the effects on arousal threshold generalize to other laboratory strains, we measured starvation-dependent changes in sleep depth in Canton-S females [33]. Again, we found that sleep is reduced and arousal threshold is increased in starved Canton-S flies (S3E–S3H Fig). Therefore, the increase in sleep depth during starvation is generalizable across sex and *D. melanogaster* strains.

It is possible that the increase in sleep depth during the night in starved flies is due to either the duration of starvation or circadian regulation. To differentiate between these possibilities, we shifted the time of starvation to the onset of lights off. In these flies, there was a small reduction in nighttime sleep and a robust decrease in daytime sleep (Fig 2K and 2L). We found that sleep depth did not differ during the night (0–12 hrs of starvation), but was significantly increased in starved flies during the day (12–24 hrs of starvation; Fig 2M and 2N). Therefore, the increase in sleep depth observed in starved flies is induced by the duration of food restriction, rather than the phase of the light cycle.

To determine whether longer periods of starvation induce recovery sleep, we extended starvation for 36 hours beginning at ZT12 (onset of lights off; Fig 3A), resulting in over 500 minutes of total sleep loss (Fig 3B and 3C). Arousal threshold was elevated from hours 12–24 and 24–36 following starvation, suggesting sleep depth is increased, even under severe starvation conditions (Fig 3D and 3E). Despite the robust loss in sleep, no change in sleep duration or arousal threshold was detected when flies were returned to food (Fig 3F and 3G). Therefore, flies do not exhibit a homeostatic rebound following prolonged periods of starvation, even though it results in significant cumulative sleep loss.

Recent work suggests energy metabolism is a critical regulator of sleep homeostasis [34], but the contributions of dietary composition to sleep homeostasis remains poorly understood. Fly food is comprised primarily of sugars and protein, with yeast providing the primary protein source [35,36]. To determine the effects of different dietary macronutrients on sleep regulation, we varied the concentration of sugar and yeast and measured the effects on sleep duration and depth. Flies were fed a diet consisting of 2% yeast alone, 2% yeast and 5% sugar, or 5% sugar alone (Fig 4A). When compared to standard food, sleep duration did not differ between any of the diets, consistent with the notion that sufficiency of total caloric content, rather than the presence of specific macronutrients, regulates sleep duration (Fig 4B and 4C; [11]). Quantification of arousal threshold revealed increased nighttime sleep depth in flies fed a sucrose-only diet, but not in flies fed yeast and sucrose, or yeast alone (Fig 4D and 4E). Given that flies increase arousal threshold on a sucrose-only diet, we next assessed whether this would have any effect during recovery. We found no change in sleep duration or arousal threshold in the subsequent 12 hours after sucrose-only feeding (Fig 4F and 4G). This suggests that even though flies fed a sucrose-only diet sleep more deeply, there is no effect on sleep duration or depth during later time periods. To specifically examine the contribution of amino acids to the regulation of arousal threshold, we supplemented sugar with a cocktail of amino

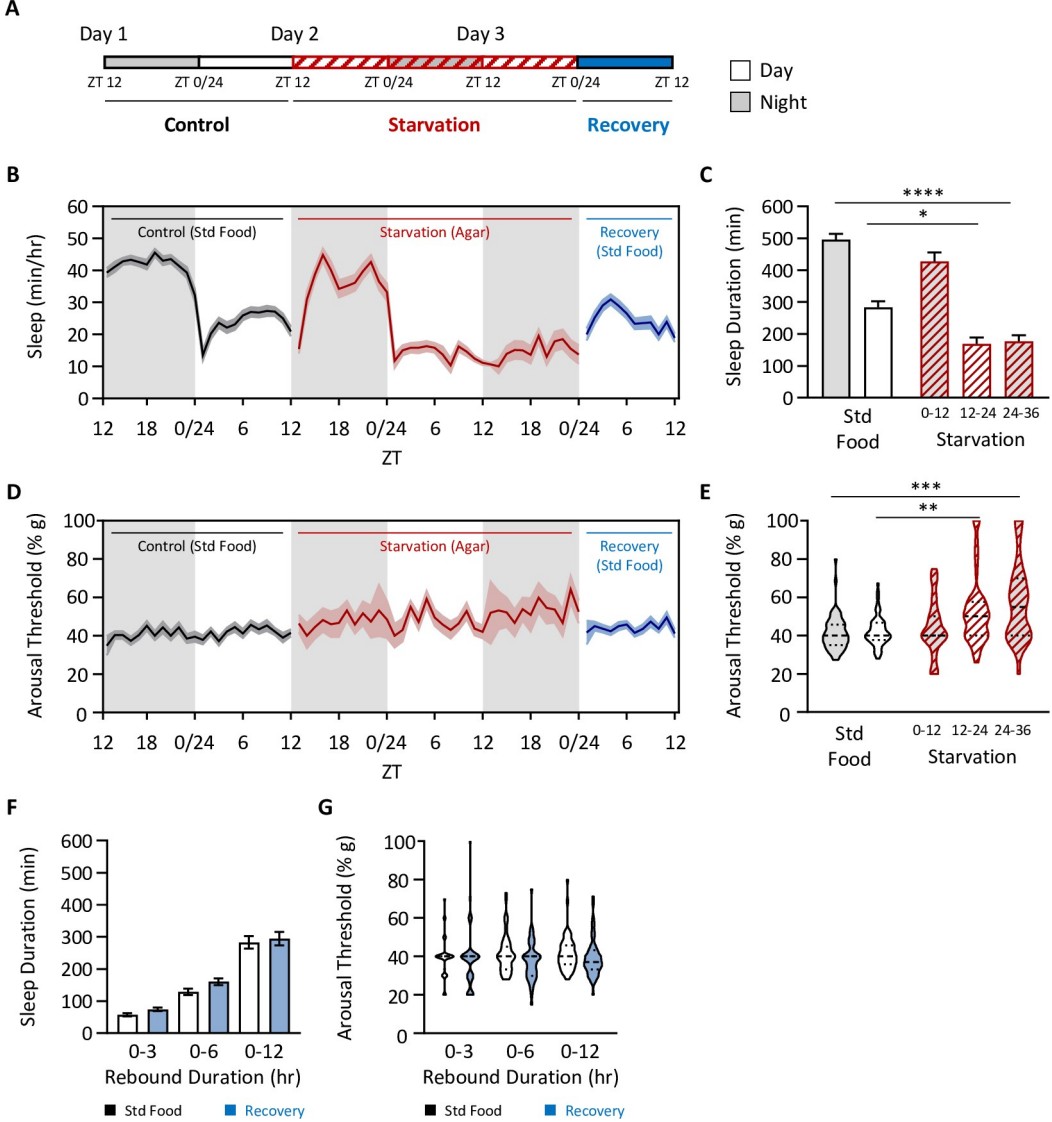

**Fig 3. The absence of a homeostatic rebound following starvation is independent of starvation duration.** (A) Total sleep and arousal threshold were assessed for 24 hrs on standard food (Control), for a subsequent 36 hrs on agar (Starvation), and then when transferred back to standard food to assess homeostatic rebound (Recovery). Flies were flipped to agar at ZT12. (B) Sleep profile. (C) Sleep duration decreases in the starved state (two-way ANOVA: $F_{4,284} = 20.44$, $P<0.0001$). *Post hoc* analyses revealed that this occurs after 12hr of starvation and is maintained over time (0–12: $P = 0.5706$; 12–24: $P = 0.0499$; 24–36: $P<0.0001$). (D) Profile of arousal threshold. (E) Arousal threshold significantly increases in the starved state (Kruskal-Wallis test: H = 29.72, $P<0.0001$; N = 57–59). *Post hoc* analyses revealed that this occurs after 12hr of starvation and is maintained over time (0–12: $P>0.9999$; 12–24: $P = 0.0006$; 24–36: $P = 0.0003$). (F-G) There is no homeostatic rebound following 36 hrs of starvation. Homeostatic rebound was measured as described in Fig 1B and was assessed in 3-, 6-, and 12-hr increments. (F) There is no difference in sleep duration (two-way ANOVA: $F_{1,336} = 3.451$, $P = 0.0641$). (G) There is no difference in arousal threshold (REML: $F_{1,108} = 3.276$, $P = 0.0731$, N = 55–59). For profiles, shaded regions indicate +/- standard error from the mean. White background indicates daytime, while gray background indicates nighttime. Error bars represent +/- standard error from the mean. For arousal threshold measurements, the median (dashed line) as well as 25th and 75th percentiles (dotted lines) are shown. *** = $P<0.001$.

acids using a previously described protocol [37]. In agreement with previous findings, sleep duration did not differ in flies fed standard food, sucrose, or sucrose and amino acids (Fig 4H and 4I). The addition of amino acids to a sucrose-only diet restored nighttime arousal threshold to the level of flies fed standard food (Fig 4J and 4K). These findings suggest the absence of dietary amino acids increases sleep depth without affecting sleep duration.

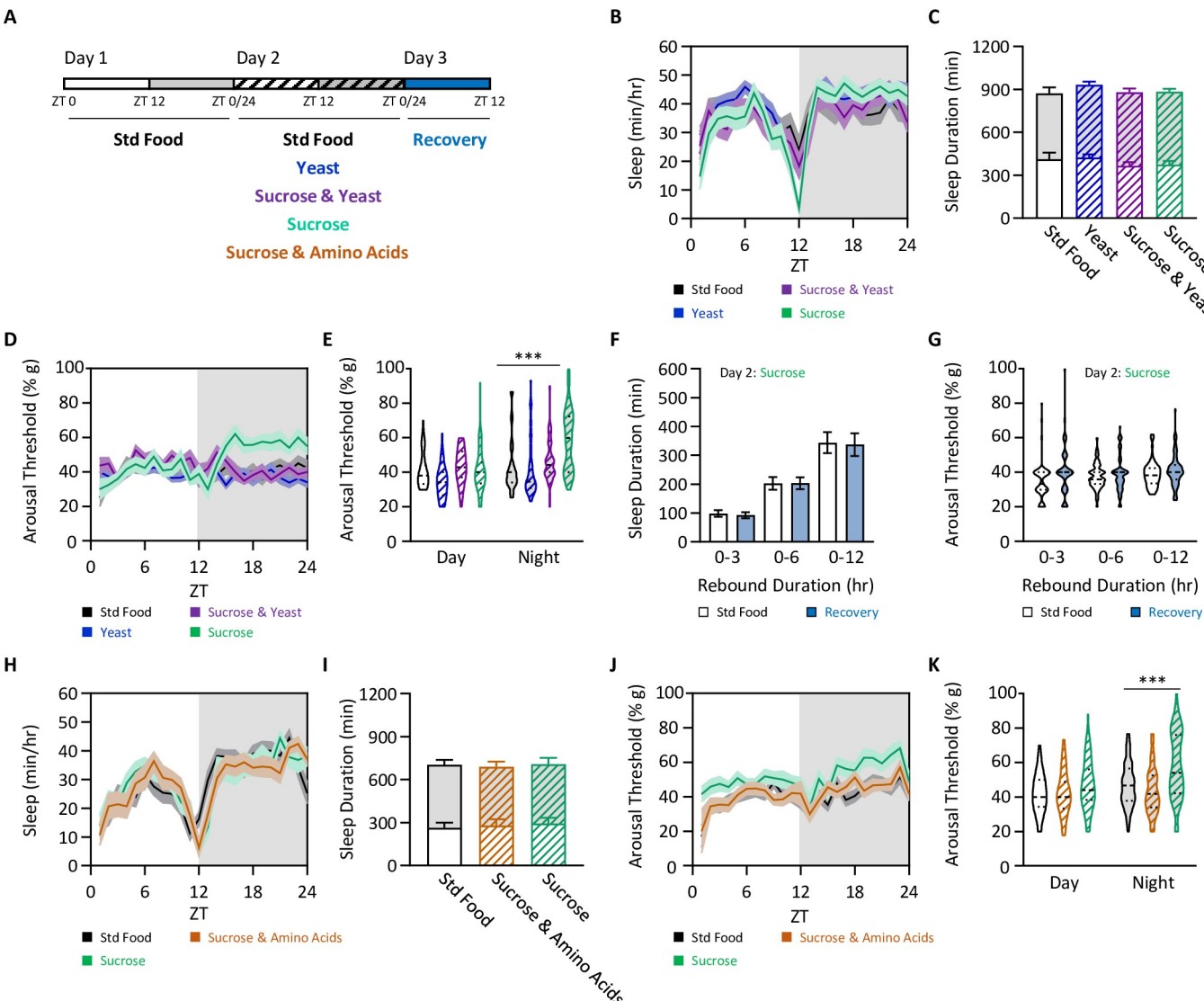

**Fig 4. The absence of dietary protein increases sleep depth.** (A) Sleep and arousal threshold measurements were taken over a 24 hr period from flies fed standard food media, 2% yeast, 2% yeast and 5% sucrose, 5% sucrose, or 5% sucrose and 2.5x amino acids. Comparisons were then made between flies fed standard food media and each of the subsequent diets. (B) Sleep profile. (C) Diet does not affect sleep duration (two-way ANOVA: $F_{1,396} = 0.3886$, $P = 0.7612$). (D) Arousal threshold profile. (E) Diet does affect nighttime sleep depth (REML: $F_{3,183} = 11.70$, $P < 0.0001$, N = 46–53). *Post hoc* analyses revealed a significant increase in nighttime arousal threshold when flies are fed 5% sucrose, compared to standard food (day: $P > 0.9999$; night: $P < 0.0001$). (F-G) There is no homeostatic rebound following 24 hrs of sucrose feeding. Homeostatic rebound was measured as described in Fig 1B and was assessed in 3-, 6-, and 12-hr increments. (F) There is no difference in sleep duration (two-way ANOVA: $F_{1,312} = 0.0344$, $P = 0.8530$). (G) There is no difference in arousal threshold (REML: $F_{88} = 2.738$, $P = 0.1005$, N = 46–53). (H-K) Adding amino acids to sucrose restores sleep depth. (H) Sleep profile. (I) Again, diet does not affect sleep duration (two-way ANOVA: $F_{2,140} = 0.0193$, $P = 0.9808$). (J) Arousal threshold profile. (K) Diet does affect arousal threshold (REML: $F_{2,180} = 11.98$, $P < 0.0001$, N = 20–33). *Post hoc* analyses revealed a significant increase in nighttime arousal threshold when flies are fed 5% sucrose, compared to standard food (day: $P = 0.1206$; night: $P < 0.0001$) and when sucrose is supplemented with amino acids (day: $P = 0.1362$; night: $P < 0.0001$). For profiles, shaded regions indicate +/- standard error from the mean. White background indicates daytime, while gray background indicates nighttime. Error bars represent +/- standard error from the mean. For arousal threshold measurements, the median (dashed line) as well as 25th and 75th percentiles (dotted lines) are shown. *** = $P < 0.001$.

To determine whether the effects of starvation can be recapitulated by inhibition of glycolysis, and thereby preventing cellular utilization of sugars and numerous dietary amino acids, we pharmacologically starved flies by feeding them standard fly food laced with the glycolysis inhibitor 2-deoxyglucose (2DG; Fig 5A; [38,39]). In agreement with our previous findings

[39], flies fed 2DG slept less than those housed on standard food (Fig 5B and 5C). Further, this decrease in sleep duration was accompanied by an increase in arousal threshold (Fig 5D and 5E), largely phenocopying starved flies. Further, there was no rebound in sleep duration or arousal threshold when 2DG-treated flies were placed back on standard food (Fig 5F and 5G), suggesting the elevated arousal threshold induced by 2DG is protective against sleep debt. Together, these findings suggest that the changes in sleep duration and arousal threshold in starved flies is a result of metabolic deprivation.

The *Drosophila* insulin producing cells (IPCs) are critical regulators of sleep and metabolic homeostasis [14,40,41], thereby raising the possibility that these cells regulate starvation-induced changes in sleep, arousal threshold, and metabolic rate. The IPCs express 3 of the 8 *Drosophila* insulin-like peptides *(*DILPS; [42,43]) and one of these, DILP2, has been previously implicated in sleep and feeding state [44,45]. To determine the effects *Dilp2* on sleep regulation, we measured sleep duration and depth in flies with disrupted *Dilp2* function. We first used RNAi targeted to *Dilp2* to selectively inhibit function within the IPCs. Immunostaining with *Dilp2* antibody confirmed that *Dilp2* levels were not detectable in experimental flies (*Dilp2*-GAL4>UAS-*Dilp2*[RNAi]; Fig 6A and 6B). Nighttime sleep duration in flies with RNAi knockdown of *Dilp2* did not differ in the fed or starved state compared to its respective control (Figs 6C and S4A). Conversely, nighttime arousal threshold was elevated in starved control flies, but did not differ between fed and starved *Dilp2*-GAL4>UAS-*Dilp2*[RNAi] flies (Fig 6D and 6E), suggesting *Dilp2* is required for starvation-dependent changes in arousal threshold. While control flies did not exhibit a rebound following deprivation, *Dilp2*-GAL4>UAS-*Dilp2*[RNAi] flies displayed a significant sleep rebound following starvation, suggesting *Dilp2* is required for increased sleep depth during starvation that likely prevents rebound (Fig 6F). No effect on arousal threshold was observed during recovery in the control nor the *Dilp2*-GAL4>UAS-*Dilp2*[RNAi] flies (S4B Fig). To validate that the effects of *Dilp2*-knockdown are not due to RNAi off-targeting, we assessed the effects of starvation on sleep depth and homeostasis in *Dilp2*[null] flies (Fig 6G–6I; [46]). Similar to RNAi knockdown flies, sleep duration was normal in fed and starved *Dilp2*[null] flies (Figs 6J and S4C), yet *Dilp2*[null] flies did not increase arousal threshold during starvation and displayed a significant rebound in sleep upon re-feeding (Fig 6K–6M). Again, no effect on arousal threshold was observed during recovery in the control nor the *Dilp2*[null] flies (S4D Fig). Further, during other manipulations of sleep deprivation, including feeding caffeine, both *Dilp2*-GAL4>UAS-*Dilp2*[RNAi] and *Dilp2*[null] flies respond similarly to the control in that they all decrease sleep duration with no change in depth during sleep deprivation, but following sleep deprivation produce a homeostatic rebound in which both sleep duration and depth increase (S5 Fig). Therefore, *Dilp2* is required for increasing nighttime sleep depth specifically during starvation, thereby circumventing the need for recovery sleep during the daytime when food is restored.

It is possible that *Dilp2* functions acutely to increase sleep depth during starvation, or that it is required during development. To differentiate between these possibilities, we utilized *Dilp2*-Geneswithch (GS) to temporally silence *Dilp2* expression in adulthood [47]. We acutely fed RU486 to flies harboring the transgene for inducible GAL4 in *Dilp2*-expressing neurons (*Dilp2*-GS-GAL4) as well as UAS-*Dilp2*[RNAi] for 24 hours prior to and during experimental manipulations (Fig 7A). We found that sleep duration in *Dilp2*-GS-GAL4>UAS-*Dilp2*[RNAi] flies fed RU486 did not differ in the fed or starved state compared to its respective controls (Figs 7B and S6A). Nighttime arousal threshold was significantly increased in all starved controls, but did not differ between fed and starved *Dilp2*-GS-GAL4>UAS-*Dilp2*[RNAi] flies fed RU486 (Fig 7C and 7D), suggesting *Dilp2* is required in adults for starvation-dependent changes in arousal threshold. While control flies did not exhibit a rebound following deprivation, *Dilp2*-GS-GAL4>UAS-*Dilp2*[RNAi] flies fed RU486 displayed a significant sleep rebound

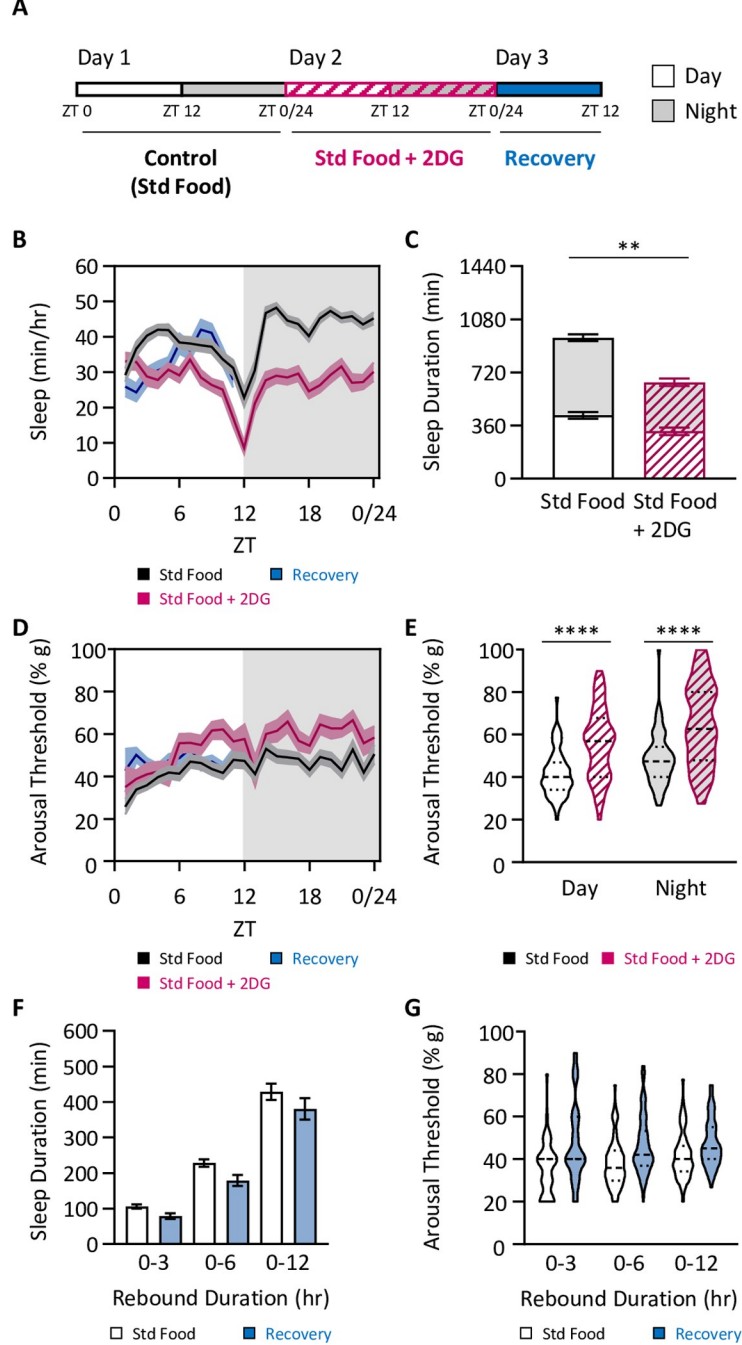

**Fig 5. Inhibition of glycolysis phenocopies the yeast-dependent modulation of arousal threshold.** (A) Total sleep and arousal threshold were assessed for 24 hrs on standard food (black outlined boxes), standard food + 2-deoxyglucose (2DG; pink outlined boxes with hatches), and then when transferred back to standard food. (B) Sleep profiles. (C) Sleep duration decreases when 2DG is included in the diet (two-way ANOVA: $F_{1,190} = 38.77$, $P<0.0001$), and occurs during both the day ($P = 0.0040$) and night ($P<0.0001$). (D) Profile of arousal threshold. (E) Arousal threshold significantly increases when 2-DG is included in the diet (REML: $F_{1,111} = 71.07$, $P<0.0001$, N = 46–51), and occurs during both the day ($P<0.0001$) and night ($P<0.0001$). (F-G) Measurements of homeostatic rebound following 24 hrs of 2DG feeding were assessed in 3-, 6-, and 12-hr increments. (F) Although there is a significant effect of treatment on sleep duration (two-way ANOVA: $F_{1,285} = 7.943$, $P = 0.0052$), *post hoc* analyses revealed no evidence of a homeostatic rebound for any timepoint measured (0–3 hrs: $P = 0.6399$; 0–6 hrs: $P = 0.1582$; 0–12 hrs: $P = 0.1661$). (G) Although there is a significant effect of treatment arousal threshold (REML: $F_{1,121} = 5.426$, $P<0.0231$, N = 46–51), *post hoc* analyses revealed no evidence of a homeostatic rebound for any timepoint measured (0–3 hrs: $P = 0.3274$; 0–6 hrs: $P = 0.0822$; 0–12 hrs: $P = 0.4670$). For profiles, shaded regions indicate +/- standard error from the mean. White

background indicates daytime, while gray background indicates nighttime. For sleep measurements, error bars represent +/- standard error from mean. For arousal threshold measurements, the median (dashed line) as well as 25th and 75th percentiles (dotted lines) are shown. ** = $P<0.01$; **** = $P<0.0001$.

following starvation, suggesting *Dilp2* is required during adulthood for increased sleep depth during starvation that likely prevents rebound (Fig 7E). No effect on arousal threshold was observed during recovery in any of the manipulations (S6B Fig). Taken together, these findings suggest *Dilp2* acts acutely to increase sleep depth during starvation and confers resiliency to starvation-induced sleep deficits.

Given that sleep depth increases not just during starvation, but also in the absence of yeast, we next tested whether *Dilp2* may similarly regulate sleep depth on a sucrose-only diet. We found that, similar to the control, feeding *Dilp2*-GAL4>UAS-*Dilp2*RNAi flies a sucrose-only diet has no effect on sleep duration (S7A Fig). However, unlike the control, sleep depth does not increase on a sucrose-only diet in *Dilp2*-GAL4>UAS-*Dilp2*RNAi flies (S7B Fig). Additionally, we did not observe any homeostatic rebound in sleep duration or depth in the control or in *Dilp2*-GAL4>UAS-*Dilp2*RNAi flies (S7C and S7D Fig), suggesting that a homeostatic rebound, similar to what we observe in these flies after 24 hours of starvation, occurs only after a decrease in both sleep duration and depth. We found similar results in *Dilp2*null flies (S7E–S7H Fig). Overall, these results suggest that *Dilp2* uniquely regulates sleep depth both during starvation and the absence of yeast.

## Discussion

Animals ranging from jellyfish to humans display a homeostatic rebound following sleep deprivation [6,48]. Here we describe a functional and molecular mechanism underlying resilience to starvation-induced sleep loss. While it is widely accepted that sleep is a homeostatically regulated process, a number of factors suggest flies may be resilient to starvation-induced changes in sleep. It has previously been reported that starvation does not induce a sleep rebound after 12 hours of starvation [22], and in appetitive conditioning assays starvation is required for memory formation, suggesting flies still form robust memories despite sleep loss [49–51]. Our findings suggest that changes in sleep quality during starvation underlie the reduced need for a rebound. This resiliency to starvation-induced sleep loss may reflect an ethologically relevant adaptation that allows animals to forage through portions of the night without requiring rebound sleep the following day.

A central model suggests the sleep homeostat is functionally distinct from circadian processes and detects an accumulation of sleep pressure [52]. While the cellular basis homeostatic regulation in response to different types of sleep loss remains poorly understood, a large-scale screen identified the anti-microbial peptide *nemuri* as a sleep-promoting factor that accumulates during periods of prolonged wakefulness [21]. In addition, numerous other genes including the TNF-alpha homolog *Eiger*, the microRNA *mir190SP*, and the rhoGTPase *Cross-veinless* are required for sleep homeostasis [53–55]. Recovery sleep following deprivation involves complex and likely redundant neural circuits [56], raising the possibility that sleep-loss inducing perturbations can differentially impact the homeostat. For example, both the fan-shaped body and ring-neurons that comprise the ellipsoid body, brain regions associated with regulation of sleep, movement, and visual memory, have been implicated in the regulation of sleep homeostasis [8,53,57,58]. In the mushroom body, activity within a subset of sleep-promoting neurons is elevated during sleep deprivation, suggesting this may provide a neural correlate of sleep drive [59]. Further, mounting evidence suggests sleep duration can be differentiated from sleep homeostasis. As such, studies using TrpA1, the heat-activated thermosensor to

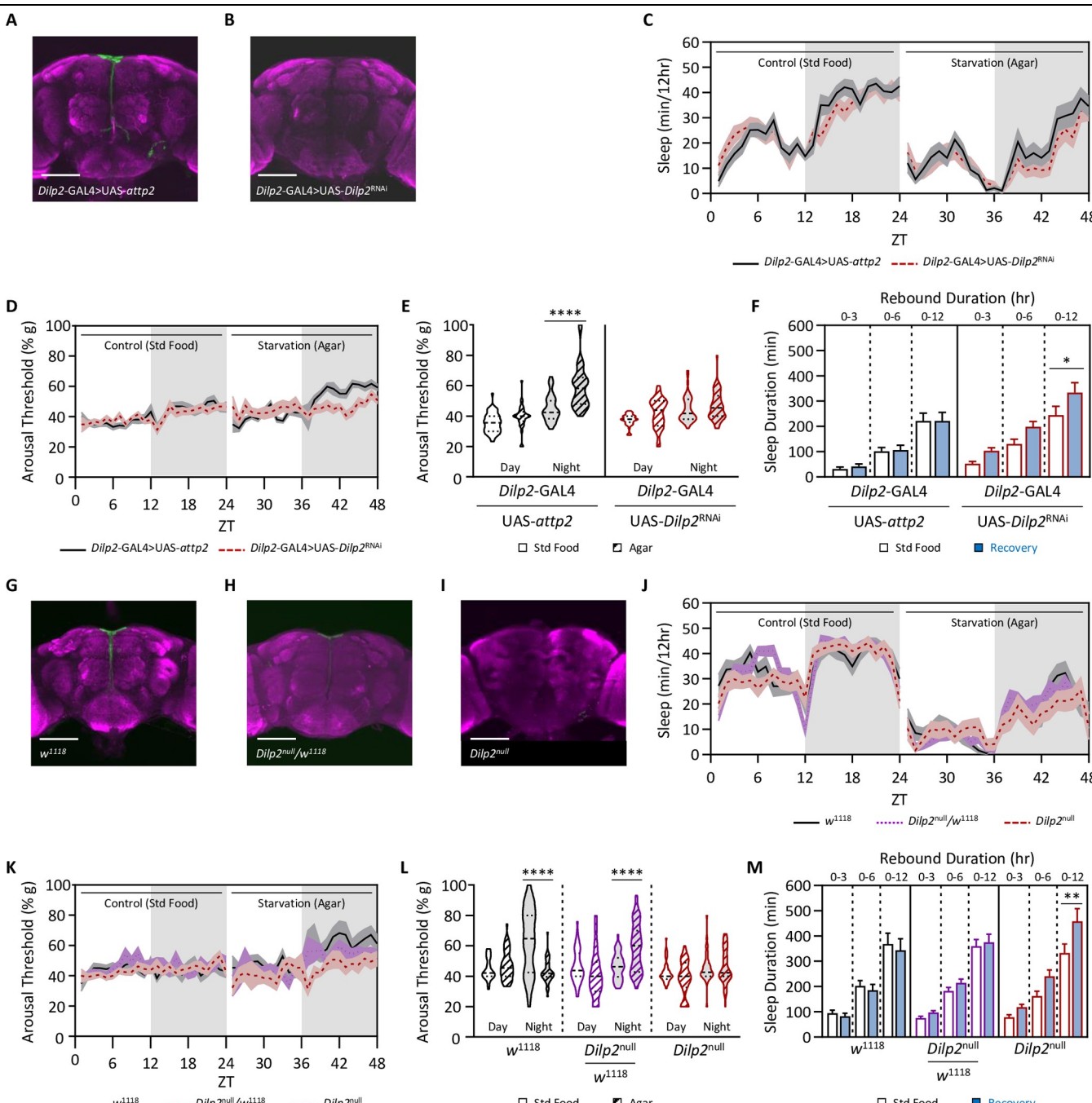

**Fig 6. *Dilp2* uniquely regulates arousal threshold during starvation.** Total sleep and arousal threshold were assessed for 24 hrs on standard food. Flies were then sleep deprived by starving them for the following 24hrs. Homeostatic rebound was then assessed during the subsequent 12 hrs. (A,B) Immunohistochemistry using *Dilp2* antibody (green). For each image, the brain was counterstained with the neuropil marker nc82 (magenta). Scale bar = 100μm. In comparison with the *Dilp2*-GAL4>UAS-*attp2* control (A), Dilp2 protein is reduced in *Dilp2*-GAL4>UAS-*Dilp2*[RNAi] (B). (C) Compared to the control (*Dilp2*-GAL4>UAS-*attp2*), knockdown of *Dilp2* in *Dilp2*-expressing neurons (*Dilp2*-GAL4>UAS-*Dilp2*[RNAi]) has no effect on nighttime sleep duration (two-way ANOVA: $F_{1,146} = 2.164$, $P = 0.1435$), however there is a significant effect of starvation (two-way ANOVA: $F_{1,146} = 26.78$, $P<0.0001$). For each genotype, post hoc analyses revealed a significant decrease in nighttime sleep duration when starved (*Dilp2*-GAL4>UAS-*attp2*: $P = 0.0008$; *Dilp2*-GAL4>UAS-*Dilp2*[RNAi]: $P = 0.0002$), when compared to standard food. (D) There is a significant effect of genotype on nighttime arousal threshold (REML: $F_{1,76} = 18.22$, $P<0.0001$). Post hoc analyses revealed that while controls significantly increase nighttime arousal threshold during starvation (*Dilp2*-GAL4>UAS-*attp2*: $P<0.0001$), there is no effect of knockdown of *Dilp2* in *Dilp2*-expressing neurons on arousal threshold (*Dilp2*-GAL4>UAS-*Dilp2*[RNAi]: $P = 0.4053$). (E-F) Measurements of homeostatic rebound following 24 hrs of starvation were assessed in 3-, 6-, and 12-hr increments. (E) There is a significant effect of genotype on sleep duration following 24 hrs of starvation (two-way ANOVA: $F_{1,146} = 8.651$, $P = 0.0038$). *Post hoc* analyses revealed that in control flies there was no change in sleep duration for any timepoint measured (*Dilp2*-GAL4>UAS-*attp2*: 0–3 hrs: $P = 0.9849$; 0–6 hrs: $P = 0.9974$; 0–12 hrs: $P>0.9999$). However, knockdown of *Dilp2* in *Dilp2*-expressing neurons significantly increases rebound sleep between 6 and 12 hrs after rebound onset (*Dilp2*-GAL4>UAS-*Dilp2*[RNAi]: 0–3 hrs: $P = 0.4030$; 0–6 hrs: $P = 0.1725$; 0–12 hrs: $P = 0.0450$). (F) Compared to the control (*Dilp2*-GAL4>UAS-*attp2*),

knockdown of *Dilp2* in *Dilp2*-expressing neurons (*Dilp2*-GAL4>UAS-*Dilp2*$^{RNAi}$) has no effect on arousal threshold following 24 hrs of starvation (REML: $F_{1,76} = 0.4683$, $P = 0.4954$). (G-I) Immunohistochemistry using DILP2 antibody (green). For each image, the brain was counterstained with the neuropil marker nc82 (magenta). Scale bar = 100μm. In comparison with the w$^{1118}$ control (H), DILP2 protein is present in heterozygotes (I), while it is absent in *Dilp2*$^{null}$ mutants (J). (J) In comparison to the control (*w*$^{1118}$), there is no effect on nighttime sleep duration in *Dilp2*$^{null}$ heterozygotes or *Dilp2*$^{null}$ mutants (two-way ANOVA: $F_{1,231} = 0.1994$, $P = 0.8194$), however there is a significant effect of starvation (two-way ANOVA: $F_{1,231} = 59.11$, $P<0.0001$). For all three genotypes, post hoc analyses revealed a significant decrease in nighttime sleep duration when starved (w$^{1118}$: $P = 0.0002$; w$^{1118}$>*Dilp2*$^{null}$: $P<0.0001$; *Dilp2*$^{null}$: $P<0.0001$). (K) There is a significant effect of genotype on nighttime arousal threshold (REML: $F_{2,117} = 10.03$, $P<0.0001$). Post hoc analyses revealed that while control flies (*w*$^{1118}$) and *Dilp2*$^{null}$ heterozygotes significantly increase nighttime arousal threshold during starvation (w$^{1118}$: $P<0.0001$; w$^{1118}$>*Dilp2*$^{null}$: $P<0.0001$), there is no effect on arousal threshold in *Dilp2*$^{null}$ mutants ($P = 0.9973$). (L-M) Measurements of homeostatic rebound following 24 hrs of starvation were assessed in 3-, 6-, and 12-hr increments. (L) There is a significant effect of genotype on sleep duration following 24 hrs of starvation (two-way ANOVA: $F_{1,231} = 3.7810$, $P = 0.0531$). *Post hoc* analyses revealed no change in sleep duration for any timepoint measured in control flies and *Dilp2* heterozygotes (*w*$^{1118}$: 0–3 hrs: $P = 0.9876$; 0–6 hrs: $P = 0.9693$; 0–12 hrs: $P = 0.9108$; w$^{1118}$>*Dilp2*$^{null}$: 0–3 hrs: $P = 0.8178$; 0–6 hrs: $P = 0.5672$; 0–12 hrs: $P = 0.9193$), however sleep duration does significantly increase in *Dilp2*$^{null}$ mutants (*Dilp2*$^{null}$: 0–3 hrs: $P = 0.7004$; 0–6 hrs: $P = 0.1719$; 0–12 hrs: $P = 0.0072$). (M) In comparison to the control (*w*$^{1118}$), there is no effect on arousal threshold following 24 hrs of starvation in *Dilp2*$^{null}$ heterozygotes or *Dilp2*$^{null}$ mutants (REML: $F_{2,117} = 1.420$, $P = 0.2347$). For sleep measurements, error bars represent +/- standard error from the mean. For arousal threshold measurements, the median (dashed line) as well as 25$^{th}$ and 75$^{th}$ percentiles (dotted lines) are shown. * = $P<0.05$; ** = $P<0.01$; *** = $P<0.001$; **** = $P<0.0001$.

activate neurons and suppress sleep identified many neuronal populations that suppressed sleep, some of which induced a rebound while others did not [9,60]. Therefore, our findings fortify the notion that different neural processes may modulate sleep homeostasis in accordance with the perturbation that induces sleep loss.

Our findings suggest the perturbation used to deprive flies of sleep is a critical factor in regulating sleep depth and homeostasis. In fruit flies, the vast majority of studies use mechanical shaking to induce sleep deprivation in flies [5,61–63]. This method is highly effective because the timing and duration of sleep loss can be precisely controlled. Here, we find that in addition to mechanical deprivation, sleep loss induced by caffeine feeding also results in sleep rebound. Our finding that flies do not rebound after starvation-induced sleep loss can be extended to other ethologically-relevant perturbations. For example, male flies deprived of sleep by pairing with a receptive female, presumably resulting in sexual excitation, does not induce a sleep rebound when the female is removed [64,65]. Together, these findings raise the possibility that flies may be resilient to some ethologically relevant forms of sleep deprivation and highlight the importance of studying the genetic and neural processes regulating sleep across diverse environmental contexts.

In mammals, sleep stages based on cortical activity are used to measure sleep depth [66]. For example, recovery sleep is marked by increases in slow wave sleep [67]. While orthologous sleep stages have not been identified in fruit flies, behavioral and physiological measurements suggests sleep can differ in intensity [24–26]. For example, sleep periods lasting longer than 15 minutes are associated with an elevated arousal threshold, suggesting that flies are in deeper sleep [24,25]. The finding that nighttime arousal threshold is elevated in starved animals suggests starvation enhances sleep depth. It is also is possible that arousal threshold will differ based on the sensory stimulus used to probe arousal. In agreement with this notion, it has previously been reported that starved flies are more readily aroused by odors, suggesting the olfactory system may be sensitized in starved animals [68]. Therefore, is likely that measuring arousal threshold using other sensory stimuli, including light, smell, or taste, may differentially impact arousal threshold.

While many studies have examined the interactions between sleep and feeding, these have typically compared fed and starved animals without examining the effects of specific dietary components on sleep regulation [11,69–71]. In *Drosophila*, feeding of a sugar only diet is sufficient for normal sleep duration [11], and evidence suggests that activation of sweet taste-receptors alone is sufficient to promote sleep [68,72]. While our group and others have interpreted this to suggest that dietary sugar alone is sufficient for 'normal sleep', these findings indicate that sleep quality

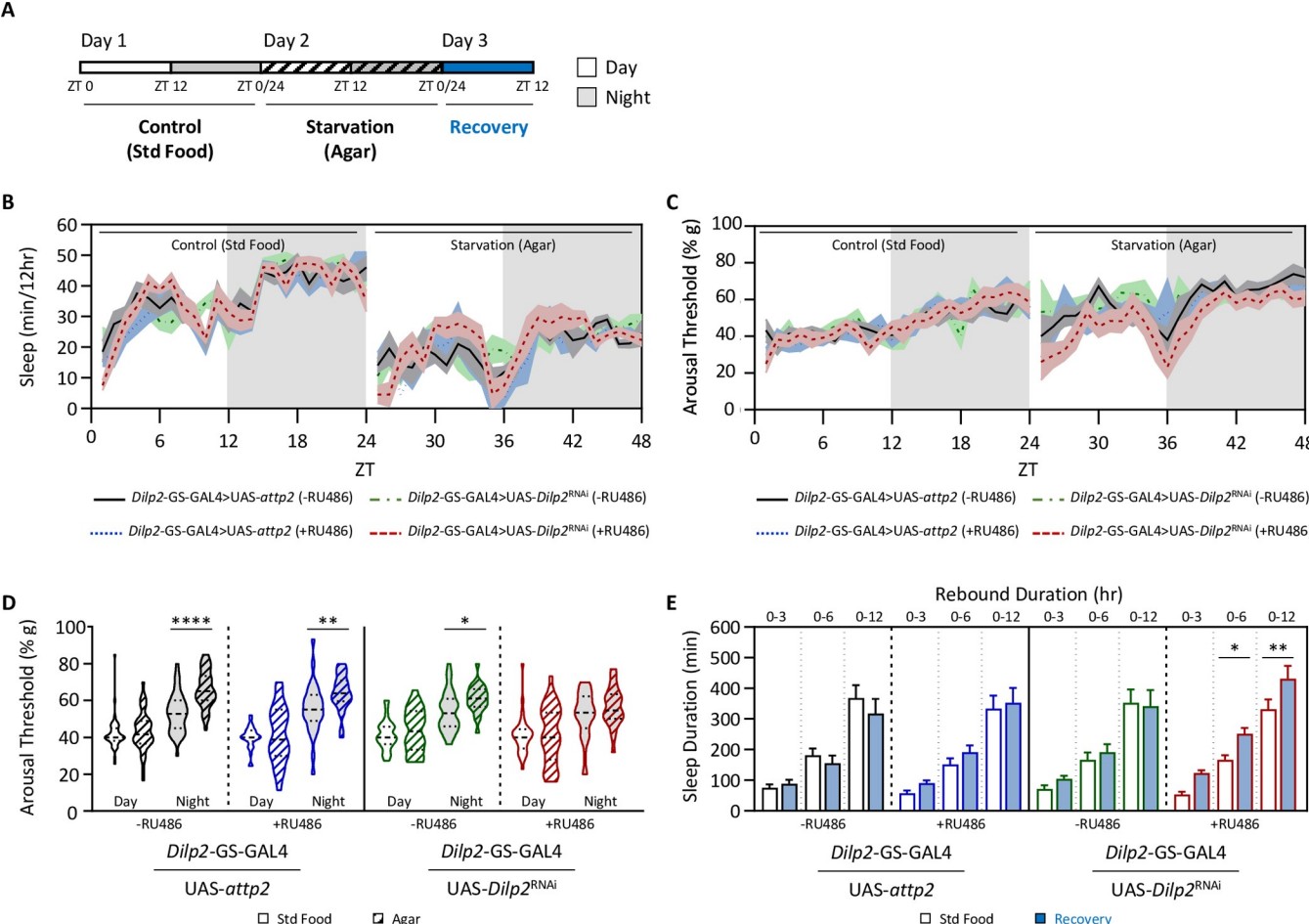

**Fig 7. *Dilp2* acutely regulates arousal threshold during starvation in adults.** (A) Silencing of *Dilp2* expression was temporally controlled using Dilp2-Geneswitch (GS). Flies were reared to adulthood and aged in the absence of RU486. Then, either solvent alone (-RU486) or RU486 (+RU486) was added to the diet 24 hrs prior to testing in the DART. (B) Total sleep and arousal threshold were assessed for 24 hrs on standard food (Day 1). Flies were then sleep deprived by starving them for the following 24hrs (Day 2). Homeostatic rebound was then assessed during the subsequent 12 hrs (Day 3). (C) There is no effect of genotype on nighttime sleep duration (two-way ANOVA: $F_{3,67} = 0.1975$, $P = 0.8977$), however there is a significant effect of starvation (two-way ANOVA: $F_{1,67} = 74.34$, $P<0.0001$). For each genotype, *post hoc* analyses revealed a significant decrease in nighttime sleep duration when starved (*Dilp2*-GS-GAL4>UAS-*attp2*/-RU486: $P = 0.0002$; *Dilp2*-GS-GAL4>UAS-*attp2*/+RU486: $P = 0.0003$; *Dilp2*-GS-GAL4>UAS-*Dilp2*^RNAi/-RU486: $P<0.0001$; *Dilp2*-GS-GAL4>UAS-*Dilp2*^RNAi/+RU486: $P = 0.006$), when compared to standard food. (D) There is a significant effect of genotype on nighttime arousal threshold (REML: $F_{1,77} = 49.98$, $P<0.0001$). *Post hoc* analyses revealed that while controls significantly increase nighttime arousal threshold during starvation (*Dilp2*-GS-GAL4>UAS-*attp2*/-RU486: $P<0.0001$; *Dilp2*-GS-GAL4>UAS-*attp2*/+RU486: $P = 0.0025$; *Dilp2*-GS-GAL4>UAS-*Dilp2*^RNAi/-RU486: $P = 0.0103$), there is no effect of knockdown of *Dilp2* in *Dilp2*-expressing neurons on arousal threshold (*Dilp2*-GS-GAL4>UAS-*Dilp2*^RNAi/+RU486: $P = 0.2989$). (E-F) Measurements of homeostatic rebound following 24 hrs of starvation were assessed in 3-, 6-, and 12-hr increments. (E) There is a significant effect of genotype on sleep duration following 24 hrs of starvation (two-way ANOVA: $F_{3,170} = 2.948$, $P = 0.0344$). *Post hoc* analyses revealed that in the controls, there were no change in sleep duration for any timepoint measured (*Dilp2*-GS-GAL4>UAS-*attp2*/-RU486: 0–3 hrs: $P = 0.9864$; 0–6 hrs: $P = 0.9022$; 0–12 hrs: $P = 0.5590$; *Dilp2*-GS-GAL4>UAS-*attp2*/+RU486: 0–3 hrs: $P = 0.9226$; 0–6 hrs: $P = 0.7724$; 0–12 hrs: $P = 0.9710$; *Dilp2*-GS-GAL4>UAS-*Dilp2*^RNAi/-RU486: 0–3 hrs: $P = 0.8609$; 0–6 hrs: $P = 0.9627$; 0–12 hrs: $P = 0.9939$). However, knockdown of *Dilp2* in *Dilp2*-expressing neurons significantly increases rebound sleep between 3 and 12 hrs after rebound onset (*Dilp2*-GS-GAL4>UAS-*Dilp2*^RNAi/+RU486: 0–3 hrs: $P = 0.1174$; 0–6 hrs: $P = 0.0395$; 0–12 hrs: $P = 0.0113$). (F) Compared to the controls, knockdown of *Dilp2* in *Dilp2*-expressing neurons (*Dilp2*-GS-GAL4>UAS-*Dilp2*^RNAi/+RU486) has no effect on arousal threshold following 24 hrs of starvation (REML: $F_{3,134} = 1.489$, $P = 0.2203$). For sleep measurements, error bars represent +/- standard error from the mean. For arousal threshold measurements, the median (dashed line) as well as 25th and 75th percentiles (dotted lines) are shown. * = $P<0.05$; ** = $P<0.01$; *** = $P<0.001$; **** = $P<0.0001$.

may differ in flies fed a sugar only-diet, and that this has impacts on sleep architecture and homeostasis.

The IPCs have long been proposed as critical integrators of behavior and metabolic function [14,40,44,73]. Our results raise the possibility that *Dilp2*-expressing neurons become

active on diets deficient in amino acids. The application of genetically encoded $Ca^{2+}$ sensors that can be measured in freely moving animals, such as CaMPARI and TRIC, provide the opportunity to determine the effects of dietary macronutrients on IPC activity [74,75]. Previous studies have implicated both the IPCs, as well as *Dilp2*, *Dilp3*, and *Dilp5* in promoting sleep during fed conditions, suggesting a role for the *Dilps* in sleep regulation [44,45]. *Dilp2*-expressing neurons are functionally downstream of wake-promoting octopamine neurons [14]. Induction of sleep loss by activating octopamine neurons does not induce a rebound [9], phenocopying loss of *Dilp2*. Together, these findings suggest a complex role for *Dilp2* and the IPCs in sleep regulation, and suggest multiple transmitters expressed in the IPCs may act in concert with *Dlip2* to differentially regulate sleep under fed and starved conditions. Additionally, we did not find any changes in sleep duration in flies lacking *Dilp2*. This is contrary to what we would predict based on previous literature, since genetic ablation of the IPCs has been shown to mimic diabetic- and starvation-like phenotypes [42,45,76]. However, possibly due to the compensatory effect of having multiple *Dilps* [46,77,78], these findings do not directly translate into expression of individual *Dilps* in the IPCs, as there are conflicting reports on whether *Dilp2* expression is modulated during starvation. Some have shown that, contrary to *Dilp3* and *Dilp5*, *Dilp2* expression does not change during starvation [42,79,80], while others have shown that *Dilp2* expression decreases [45].

Numerous factors have been identified as essential regulators of starvation-induced sleep suppression. For example, we have found that flies mutant for the mRNA/DNA binding protein *translin*, the neuropeptide *Leucokinin*, and *Astray*, a regulator of serine biosynthesis, fail to suppress sleep when starved [39,69,70]. Further, activation of orexigenic *Neuropeptide F*-expressing neurons or the sweet-sensing *Gr64f*-expressing neurons suppress sleep, suggesting that activation of feeding circuits may directly inhibit sleep [72,81]. However, it is not known whether these factors impact nutrient-dependent changes in sleep depth. A central question is how neural circuits that mediate starvation-induced sleep suppression interface with the IPCs that regulate sleep depth during the starvation period. We have found that *Leucokinin* neurons in the lateral horn signal to IPC neurons that express the *Leucokinin Receptor* (*LkR*) to suppress sleep during starvation [70]. These findings raise the possibility of a connection between LkR signaling in the IPCs and *Dilp2* function, that in turn increase sleep depth during periods of starvation.

Overall, our findings identify differential modulation of sleep duration and depth in response to ecologically relevant sleep restriction. These findings reveal that naturally occurring resilience to starvation-induced sleep loss occurs by enhancement of sleep depth. This increase in sleep depth not only occurs during starvation, but also in the absence of yeast. Our finding that *Dilp2* is required for starvation-induced modulation of sleep depth implicates insulin signaling in its regulation and highlights the need to understand the molecular mechanisms underlying changes in sleep quality in response of environmental perturbation.

## Material and methods

### Fly husbandry and stocks

Flies were grown and maintained on standard *Drosophila* food media (Bloomington Recipe, Genesee Scientific, San Diego, California) in incubators (Powers Scientific, Warminster, Pennsylvania) at 25°C on a 12:12 LD cycle with humidity set to 55–65%. The following fly strains were obtained from the Bloomington Stock Center: $w^{1118}$ (#5905); Canton-S (#64349; [33]); *Dilp2*$^{null}$ (#30881; [46]); *Dilp2*-GAL4 (#37516; [76]); UAS-*attp2* (#36303; [82]); and UAS-*Dilp2*$^{RNAi}$ (#31068; [82]). The *Dilp2*$^{null}$ flies as well as all GAL4 and UAS lines were back-crossed to the $w^{1118}$ laboratory strain for 6 generations. The Canton-S strain was used to validate that the observed effects were not specific to the $w^{1118}$ strain (S3E–S3H Fig). The *Dilp2*-

GS line was a kind gift from Dr. Heinrich Jasper [47]. All flies carrying the *Dilp2*-GS driver were placed onto standard food supplemented with either 200 μM RU486 (#M8046; Sigma, St. Louis, Missouri) to activate transgene expression or the equivalent volume of 95% ethanol (vehicle alone) for 24 hours prior to and during experimental trials. Unless otherwise stated, 3-to-5 day old mated females were used for all experiments performed in this study.

## Sleep and arousal threshold measurements

Arousal threshold was measured using the *Drosophila* Arousal Tracking system (DART), as previously described [25]. In brief, individual female flies were loaded into plastic tubes (Trikinectics, Waltham, Massachusetts) and placed onto trays containing vibrating motors. Flies were recorded continuously using a USB-webcam (QuickCam Pro 900, Logitech, Lausanne, Switzerland) with a resolution of 960x720 at 5 frames per second. The vibrational stimulus, video tracking parameters, and data analysis were performed using the DART interface developed in MATLAB (MathWorks, Natick, Massachusetts). To track fly movement, raw video flies were subsampled to 1 frame per second. Fly movement, or a difference in pixilation from one frame to the next, was detected by subtracting a background image from the current frame. The background image was generated as the average of 20 randomly selected frames from a given video. Fly activity was measured as movement of greater than 3 mm. Sleep was determined by the absolute location of each fly and was measured as bouts of immobility for 5 min or more. Arousal threshold was assessed using sequentially increasing vibration intensities, from 0 to 1.2 g, in 0.3 g increments, with an inter-stimulus delay of 15 s, once per hour over 24 hours starting at ZT0. Measurements of arousal threshold are reported as the proportion of the maximum force applied to the platform, thus an arousal threshold of 40% is 40% of 1.2g. For experiments using the *Drosophila* Activity Monitoring (DAM) system (Trikinetics, Waltham, MA, USA), measurements of sleep and waking activity were measured as previously described [61,83]. For each individual fly, the DAM system measures activity by counting the number of infrared beam crossings over time. These activity data were then used to calculate sleep, defined as bouts of immobility of 5 min or more. Sleep traits were then extracted using the *Drosophila* Sleep Counting Macro [27].

## Sleep deprivation

For all sleep deprivation experiments, flies were briefly anesthetized using $CO_2$ and then placed into plastic tubes containing standard food. Flies were allowed to recover from $CO_2$ exposure and to acclimate to the tubes for a minimum of 24 hours prior to the start of the experiment. Upon experiment onset, baseline sleep and arousal threshold were measured starting at ZT0 for 24 hours. Unless otherwise stated, for the following 24 hours, flies were sleep deprived using one of the methods described below, during which sleep and arousal threshold were also measured. Lastly, to assess homeostatic rebound, flies were returned to standard conditions and sleep and arousal threshold was measured during the subsequent day (ZT0-ZT12). To assess the effect of sleep deprivation on sleep duration and depth, baseline nighttime sleep and arousal threshold were compared with nighttime sleep and arousal threshold measurements during sleep deprivation. To determine whether there exists a homeostatic rebound in sleep duration and/or depth, baseline daytime sleep and arousal threshold were compared to daytime sleep and arousal threshold during recovery.

**Caffeine.** Caffeine (#C0750, Sigma) was added to melted standard *Drosophila* media at a concentration of 0.5 mg/mL. At ZT0 on the day of sleep deprivation, flies were transferred into tubes containing the caffeine-laced standard food. Following 24 hours of caffeine

supplementation, flies were then transferred back into tubes containing standard *Drosophila* media during the recovery period.

**Mechanical vibration.** For mechanical vibration experiments, the motors of the DART system were utilized to disrupt sleep, as described previously [25]. Briefly, a randomized mechanical stimulus, set at a maximum of 1.2 g, occurred every 20–60 sec, with each stimulus composing 4–7 pulses lasting 0.5–4 sec in length. These randomized stimuli began at ZT0 on the day of sleep deprivation and was repeated for 24hours.

**Starvation.** Flies were starved on 1% agar (#BP1423, Fisher Scientific, Hampton, New Hampshire), a non-nutritive substrate. At ZT0 on the day of sleep deprivation, flies were transferred into tubes containing agar. Following 24 hours of starvation, flies were then transferred back into tubes containing standard *Drosophila* media during the recovery period.

**Pharmacological manipulations.** 2-Deoxyglucose (2DG; #D8375, Sigma) was added to melted standard *Drosophila* media at a concentration of 400 mM, as used previously to suppress sleep [39]. At ZT0 on the day of sleep deprivation, flies were transferred into tubes containing the 2DG-laced standard food media. Following 24 hours of 2DG supplementation, flies were then transferred back into tubes containing standard *Drosophila* media during the recovery period.

## Feeding treatments

To assess sleep and arousal threshold on different diets, a similar experimental design was employed as that described above. In lieu of sleep deprivation, sleep duration and depth were assessed on one of four different food treatments. Flies were anesthetized using $CO_2$ and then placed into plastic tubes containing standard food. Flies were allowed to recover from $CO_2$ exposure and acclimate to the tubes for a minimum of 24 hours prior to the start of the experiment. Upon experiment onset, baseline sleep and arousal threshold were measured starting at ZT0 for 24 hours. For the following 24 hours, flies were transferred into tubes containing one of five different diets, during which sleep and arousal threshold were again measured. These diets include standard food, 2% yeast extract (#BP1422, Fisher Scientific), 2% yeast extract and 5% sucrose (#S3, Fisher Scientific), 5% sucrose alone, or 5% sucrose and 2.5x amino acids (#11140050 and #1113051; ThermoFisher Scientific, Waltham, Massachusetts). With the exception of standard food, all diets were dissolved in ddH2O and included 1% agar, 0.375% tegosept (methyl-4-hydroxybenzoate dissolved in 95% ethanol; #W271004; Sigma), and 0.125% penicillin (#15140122, ThermoFisher Scientific).

## Immunohistochemistry

Brains of three to five day old female flies were dissected in ice-cold phosphate buffered saline (PBS) and fixed in 4% formaldehyde, PBS, and 0.5% Triton-X for 35 min at room temperature, as previously described [84]. Brains were then rinsed 3x with cold PBS and 0.5% Triton-X (PBST) for 10 min at room temperature and then overnight at 4˚C. The following day, the brains were incubated for 24 hours in primary antibody (1:20 mouse nc82; Iowa Hybridoma Bank; The Developmental Studies Hybridoma Bank, Iowa City, Iowa), and then diluted in 0.5% PBST at 4˚C on a rotator. The brains were incubated for another 24 hours at 4˚C upon the addition of a second primary antibody (1:1300 rabbit anti-*Dilp2*). The following day, the brains were rinsed 3x in cold PBST for 10 min at room temperature and then incubated in secondary antibody (1:400 donkey anti-rabbit Alexa 488 and 1:400 donkey anti-mouse Alexa 647; ThermoFisher Scientific, Waltham, Massachusetts) for 95 min at room temperature. The brains were again rinsed 3x in cold PBST for 10 min at room temperature then stored overnight in 0.5% PBST at 4˚C. Lastly, the brains were mounted in Vectashield (VECTOR

Laboratories, Burlingame, California) and imaged in 2μm sections on a Nikon A1R confocal microscope (Nikon, Tokyo, Japan) using a 20X oil immersion objective. Images are presented as the Z-stack projection through the entire brain and processed using ImageJ2.

## Statistical analysis

Measurements of sleep duration are presented as bar graphs displaying the mean ± standard error. Unless otherwise noted, a one-way or two-way analysis of variance (ANOVA) was used for comparisons between two or more genotypes and one treatment or two or more genotypes and two treatments, respectively. Measurements of arousal threshold were not normally distributed and so are presented as violin plots; indicating the median, $25^{th}$, and $75^{th}$ percentiles. The non-parametric Kruskal-Wallis test was used to compare two or more genotypes. To compare two or more genotypes and two treatments, a restricted maximum likelihood (REML) estimation was used. All *post hoc* analyses were performed using Sidak's multiple comparisons test. All statistical analyses were performed using InStat software (GraphPad Software 8.0).

## Supporting information

**S1 Fig. Arousal threshold does not increase as a result of acclimation to the mechanical stimulus.** (A) Total sleep and arousal threshold were assessed on standard food over a 3-day period. (B) Sleep profile. (C) There is no change in sleep duration over a 3-day period (two-way ANOVA: $F_{2,168} = 0.24$, $P < 0.79$). This is consistent during the day (ANOVA: $F_{2,84} = 0.03$, $P < 0.96$) and night (ANOVA: $F_{2,84} = 0.35$, $P < 0.70$). (D) Profile of arousal threshold. (E) There is no change in arousal threshold over a 3-day period (REML: $F_{2,56} = 0.17$, $P < 0.83$). This is consistent during the day (Kruskal-Wallis test: $H = 0.09$, $P < 0.95$; N = 29) and the night (Kruskal-Wallis test: $H = 0.23$, $P < 0.89$; N = 29). For profiles, shaded regions indicate +/- standard error from the mean. White background indicates daytime, while gray background indicates nighttime. For sleep measurements, error bars represent +/- standard error from the mean. For arousal threshold measurements, the median (dashed line) as well as $25^{th}$ and $75^{th}$ percentiles (dotted lines) are shown. (PDF)

**S2 Fig. Sleep consolidation during sleep deprivation is treatment dependent.** (A) Sleep traits were measured using the *Drosophila* Activity Monitoring (DAM) system. Individual flies were placed inside plastic tubes containing standard food at one end and a foam plug at the other. Flies were then allowed to acclimate for 24 hrs. (B) At ZT0, baseline sleep was measured on a standard food diet (Day 1). At ZT0 of the following day, flies were sleep deprived using 0.5mg/mL caffeine, mechanical vibration, or starved for 24 hrs. (C) Total sleep duration significantly decreases when caffeine is added to a standard food diet (t-test: $t_{64} = 6.54$, $P < 0.0001$). (D,E) This is not due to a change in bout number (t-test: $t_{64} = 0.24$, $P < 0.80$), but rather a significant decrease in bout length (t-test: $t_{64} = 4.82$, $P < 0.0001$). (F) Total sleep duration significantly decreases when flies are mechanically sleep deprived (t-test: $t_{56} = 12.06$, $P < 0.0001$). (G, H) This is caused by a significant decrease in bout number (t-test: $t_{56} = 7.13$, $P < 0.0001$) as well as a significant decrease in bout length (t-test: $t_{54} = 3.24$, $P < 0.0020$). (I) Total sleep duration significantly decreases during starvation (t-test: $t_{68} = 9.69$, $P < 0.0001$). (J,K) This is due to a significant decrease in bout number (t-test: $t_{68} = 11.85$, $P < 0.0001$), but not bout length (t-test: $t_{68} = 1.22$, $P < 0.22$). Error bars represent +/- standard error from the mean. ** = $P < 0.01$; **** = $P < 0.0001$. (PDF)

**S3 Fig. Starvation increases arousal threshold in males as well as in independent laboratory strain.** All experiments were performed as described in Fig 2A. (A-D) Sleep and arousal

threshold measurements in $w^{1118}$ male flies. (A) Sleep profiles of fed and starved flies. (B) Sleep duration decreases in the starved state (two-way ANOVA: $F_{1,152}$ = 101.4, $P < 0.0001$), and occurs during both the day ($P < 0.0001$) and night ($P < 0.0001$). (C) Profile of arousal threshold of fed and starved flies. (D) Arousal threshold significantly increases in the starved state (REML: $F_{1,71}$ = 40.81, $P < 0.0001$), but occurs only at night (day: $P < 0.11$; night: $P < 0.0001$). (E-H) Sleep and arousal threshold measurements in female Canton-S flies. (E) Sleep profiles of fed and starved flies. (F) Sleep duration decreases in the starved state (two-way ANOVA: $F_{1,122}$ = 36.92, $P < 0.0001$), and occurs during both the day ($P < 0.0001$) and night ($P < 0.0001$). (G) Profile of arousal threshold of fed and starved flies. (H) Arousal threshold significantly increases in the starved state (REML: $F_{1,52}$ = 62.11, $P < 0.0001$), and occurs both during the day and at night (day: $P < 0.0008$; night: $P < 0.0001$). For profiles, shaded regions indicate +/- standard error from the mean. White background indicates daytime, while gray background indicates nighttime. For sleep measurements, error bars represent +/- standard error from the mean. For arousal threshold measurements, the median (dashed line) as well as 25th and 75th percentiles (dotted lines) are shown. *** = $P < 0.001$; **** = $P < 0.0001$.
(PDF)

**S4 Fig. *Dilp2* does not regulate starvation-induced sleep suppression or arousal threshold during recovery.** (A) Compared to the control, knockdown of *Dilp2* in *Dilp2*-expressing does not affect sleep duration (two-way ANOVA: $F_{1,146}$ = 2.16, $P < 0.14$), and both genotypes suppressed sleep during starvation (two-way ANOVA: $F_{1,146}$ = 26.78, $P < 0.0001$). For each genotype, post hoc analyses revealed a significant decrease in sleep duration when starved both during the day (*Dilp2*-GAL4>UAS-*attp2*: $P < 0.03$; *Dilp2*-GAL4>UAS-*Dilp2*^RNAi^: $P < 0.0048$) and night (*Dilp2*-GAL4>UAS-*attp2*: $P < 0.0008$; *Dilp2*-GAL4>UAS-*Dilp2*^RNAi^: $P < 0.0002$). (B) Compared to the control, knockdown of *Dilp2* in *Dilp2*-expressing neurons has no effect on arousal threshold during recovery (REML: $F_{1,76}$ = 0.46, $P < 0.49$). (C) In comparison to the control, there is no effect on sleep duration in heterozygotes or *Dilp2*^null^ flies (two-way ANOVA: $F_{1,231}$ = 0.19, $P < 0.81$), and all genotypes suppressed sleep during starvation (two-way ANOVA: $F_{1,231}$ = 59.11, $P < 0.0001$). For all three genotypes, post hoc analyses revealed a significant decrease in sleep duration when starved both during the day ($w^{1118}$: $P < 0.0001$; $w^{1118}$/*Dilp2*^null^: $P < 0.0001$; *Dilp2*^null^: $P < 0.0001$) and night ($w^{1118}$: $P < 0.0002$; $w^{1118}$/*Dilp2*^null^: $P < 0.0001$; *Dilp2*^null^: $P < 0.0001$). (D) In comparison to the control, there is no effect on arousal threshold during recovery in heterozygotes or *Dilp2*^null^ flies (REML: $F_{2,117}$ = 1.42, $P < 0.23$). For sleep measurements, error bars represent +/- standard error from the mean. For arousal threshold measurements, the median (dashed line) as well as 25th and 75th percentiles (dotted lines) are shown. Measurements of homeostatic rebound were assessed in 3-, 6-, and 12-hr increments. * = $P < 0.05$; ** = $P < 0.01$; *** = $P < 0.001$; **** = $P < 0.0001$.
(PDF)

**S5 Fig. *Dilp2* does not regulate homeostatic rebound following other methods of sleep deprivation.** Total sleep and arousal threshold during sleep deprivation and recovery were assessed as described in Fig 1A. Flies were sleep deprived by adding 0.5mg/mL caffeine to their diet. (A) There is no effect of genotype on nighttime sleep duration (two-way ANOVA: $F_{1,154}$ = 0.02, $P < 0.87$), and all genotypes suppressed sleep when fed caffeine (two-way ANOVA: $F_{1,154}$ = 20.56, $P < 0.0001$). Post hoc analyses revealed a significant decrease in nighttime sleep duration when fed caffeine for both genotypes (*Dilp2*-GAL4>UAS-*attp2*: $P < 0.0069$; *Dilp2*--GAL4>UAS-*Dilp2*^RNAi^: $P < 0.0015$). (B) Similar to the control, knockdown of *Dilp2* in *Dilp2*-expressing neurons has no effect on nighttime arousal threshold when fed caffeine (REML: $F_{1,78}$ = 2.22, $P < 0.13$). (C) There is no effect of genotype on sleep duration (two-way ANOVA: $F_{1,154}$ = 0.13, $P < 0.71$), however there was a significant effect of recovery (two-way ANOVA:

$F_{1,154} = 15.80$, $P<0.0001$). For both genotypes, post hoc analyses revealed a significant increase in sleep duration after 6 hrs of recovery (*Dilp2*-GAL4>UAS-*attp2*: 0–3 hrs: $P<0.90$; 0–6 hrs: $P<0.34$; 0–12 hrs: $P<0.0005$; *Dilp2*-GAL4>UAS-*Dilp2*$^{RNAi}$: 0–3 hrs: $P<0.99$; 0–6 hrs: $P<0.94$; 0–12 hrs: $P<0.03$). (D) There was no effect of genotype on arousal threshold (REML: $F_{1,78} = 0.06$, $P<0.79$), however there was a significant effect of recovery (REML: $F_{1,76} = 106.1$, $P<0.0001$). For all both genotypes, post hoc analyses revealed a significant increase in arousal threshold after 3 hrs of recovery (*Dilp2*-GAL4>UAS-*attp2*: 0–3 hrs: $P<0.05$; 0–6 hrs: $P<0.0001$; 0–12 hrs: $P<0.0001$; *Dilp2*-GAL4>UAS-*Dilp2*$^{RNAi}$: 0–3 hrs: $P<0.09$; 0–6 hrs: $P<0.0064$; 0–12 hrs: $P<0.0012$). (E) There was no effect of genotype on nighttime sleep duration (two-way ANOVA: $F_{2,208} = 0.11$, $P<0.88$), and all genotypes suppressed sleep when fed caffeine (two-way ANOVA: $F_{2,208} = 45.32$, $P<0.0001$). Post hoc analyses revealed a significant decrease in nighttime sleep duration when fed caffeine for all three genotypes ($w^{1118}$: $P<0.0016$; $w^{1118}$/*Dilp2*$^{null}$: $P<0.0001$; *Dilp2*$^{null}$: $P<0.0003$). (F) Similar to control flies, there is no change in nighttime arousal threshold when fed caffeine in heterozygotes or *Dilp2*$^{null}$ flies (REML: $F_{2,105} = 2.74$, $P<0.06$). (G) There was no effect of genotype on sleep duration (two-way ANOVA: $F_{1,208} = 0.13$, $P<0.87$), but there was a significant effect of recovery (two-way ANOVA: $F_{1,208} = 25.08$, $P<0.0001$). Post hoc analyses revealed a significant increase in sleep duration during recovery for all three genotypes ($w^{1118}$: 0–3 hrs: $P<0.33$; 0–6 hrs: $P<0.0051$; 0–12 hrs: $P<0.0001$; $w^{1118}$/*Dilp2*$^{null}$: 0–3 hrs: $P<0.76$; 0–6 hrs: $P<0.04$; 0–12 hrs: $P<0.0060$; *Dilp2*$^{null}$: 0–3 hrs: $P<0.80$; 0–6 hrs: $P<0.02$; 0–12 hrs: $P<0.0032$). (H) There was no effect of genotype on arousal threshold (REML: $F_{2,105} = 0.370$, $P<0.68$), however there was a significant effect of recovery (REML: $F_{1,103} = 29.33$, $P<0.0001$). Post hoc analyses revealed a significant increase in arousal threshold during recovery for all three genotypes ($w^{1118}$: 0–3 hrs: $P<0.33$; 0–6 hrs: $P<0.0002$; 0–12 hrs: $P<0.0001$; $w^{1118}$/*Dilp2*$^{null}$: 0–3 hrs: $P<0.99$; 0–6 hrs: $P<0.02$; 0–12 hrs: $P<0.0026$; *Dilp2*$^{null}$: 0–3 hrs: $P<0.18$; 0–6 hrs: $P<0.01$; 0–12 hrs: $P<0.0049$). For sleep measurements, error bars represent +/- standard error from the mean. For arousal threshold measurements, the median (dashed line) as well as 25th and 75th percentiles (dotted lines) are shown. Measurements of homeostatic rebound during recovery were assessed in 3-, 6-, and 12-hr increments. * = $P<0.05$; ** = $P<0.01$; *** = $P<0.001$; **** = $P<0.0001$. (PDF)

**S6 Fig. Acute regulation of *Dilp2* expression has no effect on starvation-induced sleep suppression or arousal threshold during recovery.** (A) There is no effect of genotype on nighttime sleep duration (two-way ANOVA: $F_{3,67} = 0.19$, $P<0.89$), and all genotypes suppressed sleep during starvation (two-way ANOVA: $F_{1,67} = 74.34$, $P<0.0001$). Post hoc analyses revealed a significant decrease in sleep duration when starved both during the day (*Dilp2*-GS-GAL4>UAS-*attp2*/-RU486: $P<0.0048$; *Dilp2*-GS-GAL4>UAS-*attp2*/+RU486: $P<0.04$; *Dilp2*-GS-GAL4>UAS-*Dilp2*$^{RNAi}$/-RU486: $P<0.04$; *Dilp2*-GS-GAL4>UAS-*Dilp2*$^{RNAi}$/+RU486: $P<0.02$) and night (*Dilp2*-GS-GAL4>UAS-*attp2*/-RU486: $P<0.0002$; *Dilp2*-GS-GAL4>UAS-*attp2*/+RU486: $P<0.0003$; *Dilp2*-GS-GAL4>UAS-*Dilp2*$^{RNAi}$/-RU486: $P<0.0001$; *Dilp2*-GS-GAL4>UAS-*Dilp2*$^{RNAi}$/+RU486: $P<0.006$). (B) Compared to the controls, knockdown of *Dilp2* in *Dilp2*-expressing neurons (*Dilp2*-GS-GAL4>UAS-*Dilp2*$^{RNAi}$/+RU486) has no effect on arousal threshold following 24 hrs of starvation (REML: $F_{3,134} = 1.48$, $P<0.22$). For sleep measurements, error bars represent +/- standard error from the mean. For arousal threshold measurements, the median (dashed line) as well as 25th and 75th percentiles (dotted lines) are shown. Measurements of homeostatic rebound were assessed in 3-, 6-, and 12-hr increments. * = $P<0.05$; ** = $P<0.01$; *** = $P<0.001$; **** = $P<0.0001$. (PDF)

**S7 Fig. *Dilp2* uniquely regulates the yeast-dependent modulation of arousal threshold.**
Total sleep and arousal threshold were assessed as described in Fig 4A. On Day 2 of testing,
flies were fed a diet of 5% sucrose. (A) Compared to the control, knockdown of *Dilp2* in *Dilp2*-
expressing neurons has no effect on sleep duration when fed a sucrose-only diet (two-way
ANOVA: $F_{1,150} = 0.21$, $P<0.64$). (B) There is a significant effect of genotype on nighttime
arousal threshold (REML: $F_{1,75} = 26.51$, $P<0.0001$). *Post hoc* analyses revealed that while con-
trols significantly increase nighttime arousal threshold when fed a sucrose-only diet
($P<0.0001$), there is no effect upon knockdown of *Dilp2* in *Dilp2*-expressing neurons
($P<0.13$). (C) Similar to the control, knockdown of *Dilp2* in *Dilp2*-expressing neurons does
not change sleep duration during recovery when fed a sucrose-only diet (two-way ANOVA:
$F_{1,150} = 0.19$, $P<0.66$). (D) There is a significant effect of genotype on arousal threshold during
recovery (REML: $F_{1,75} = 6.21$, $P<0.01$). However, post hoc analyses revealed no differences in
arousal threshold in the control ($P<0.20$), nor upon knockdown of *Dilp2* in *Dilp2*-expressing
neurons ($P<0.12$). (E) In comparison to the control, there is no effect on nighttime sleep dura-
tion in heterozygotes or *Dilp2*$^{null}$ flies when fed a sucrose-only diet (two-way ANOVA: $F_{2,216} =$
$0.02$, $P<0.97$). (F) There is a significant effect of genotype on nighttime arousal threshold
(REML: $F_{2,108} = 5.93$, $P<0.0032$). Post hoc analyses revealed that while controls and heterozy-
gotes significantly increase nighttime arousal threshold when fed a diet of sucrose ($w^{1118}$:
$P<0.0001$; $w^{1118}$/*Dilp2*$^{null}$: $P<0.0001$), there is no effect on arousal threshold in *Dilp2*$^{null}$ flies
($P<0.10$). (G) Similar to the control, there is no effect on sleep duration during recovery in
heterozygotes or *Dilp2*$^{null}$ flies (two-way ANOVA: $F_{2,216} = 0.01$, $P<0.98$). (H) There is a signifi-
cant effect of genotype on arousal threshold during recovery (REML: $F_{2,108} = 14.58$,
$P<0.0001$). However, post hoc analyses revealed no differences in arousal threshold in control
flies ($P<0.67$), heterozygotes ($P<0.14$), nor *Dilp2*$^{null}$ flies ($P<0.94$). For sleep measurements,
error bars represent +/- standard error from the mean. For arousal threshold measurements,
the median (dashed line) as well as 25$^{th}$ and 75$^{th}$ percentiles (dotted lines) are shown. Measure-
ments of homeostatic rebound were assessed in 3-, 6-, and 12-hr increments. **** = $P<0.0001$.
(PDF)

**S1 Data.**
(XLSX)

## Acknowledgments

We are thankful to members of the Keene laboratory for helpful discussions and technical sup-
port. We are also thankful to Jan Veenstra for kindly providing anti-*Dilp2* antibody and Dra-
gana Rogulja for helpful discussion.

## Author Contributions

**Conceptualization:** Elizabeth B. Brown, Richard Faville, Alex C. Keene.

**Data curation:** Elizabeth B. Brown, Richard Faville.

**Formal analysis:** Elizabeth B. Brown, Kreesha D. Shah.

**Funding acquisition:** Alex C. Keene.

**Investigation:** Elizabeth B. Brown, Kreesha D. Shah.

**Methodology:** Elizabeth B. Brown, Benjamin Kottler, Alex C. Keene.

**Project administration:** Alex C. Keene.

**Resources:** Richard Faville, Benjamin Kottler, Alex C. Keene.

**Software:** Richard Faville, Benjamin Kottler.

**Supervision:** Alex C. Keene.

**Validation:** Elizabeth B. Brown.

**Visualization:** Elizabeth B. Brown.

**Writing – original draft:** Elizabeth B. Brown, Alex C. Keene.

**Writing – review & editing:** Elizabeth B. Brown, Kreesha D. Shah, Richard Faville, Benjamin Kottler, Alex C. Keene.

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
