## [Decision Letter · Decision Letter 0]

2 Aug 2019

Dear Dr Keene,

Thank you very much for submitting your Research Article entitled 'Drosophila insulin-like peptide 2 mediates dietary regulation of sleep intensity' to PLOS Genetics. Your manuscript was fully evaluated at the editorial level and by independent peer reviewers. The reviewers appreciated the attention to an important problem, but raised some substantial concerns about the current manuscript. Based on the reviews, we will not be able to accept this version of the manuscript, but we would be willing to review again a much-revised version. We cannot, of course, promise publication at that time.

All three reviewers and I agree that this manuscript addresses interesting questions about how different types of sleep loss influence sleep homeostasis and about how different nutrients modulate such relationships. In my opinion, the scope and subject are appropriate for the readers of PLOS Genetics. However, reviewers 1 and 3 list several major concerns that must be addressed before the manuscript can be considered further for publication. Some of these will require new data and significant revision to the manuscript structure. These include but are not limited to: (i) changes in how sleep and sleep rebound are measured and presented to ensure an appropriate time frame and (ii) additional evidence and mechanistic insight into role of Dilp2 signaling in modulating sleep intensity and rebound following starvation. If the authors believe that they can address these concerns in full, I would recommend a significant revision and resubmission, at which point I would ask the reviewers to evaluate the revised manuscript.

If you decide to revise the manuscript for further consideration at PLOS Genetics, please aim to resubmit within the next 60 days, unless it will take extra time to address the concerns of the reviewers, in which case we would appreciate an expected resubmission date by email to plosgenetics@plos.org.

[LINK]

We are sorry that we cannot be more positive about your manuscript at this stage. Please do not hesitate to contact us if you have any concerns or questions.

Yours sincerely,

Scott D Pletcher

Guest Editor

PLOS Genetics

Gregory P. Copenhaver

Editor-in-Chief

PLOS Genetics

Reviewer's Responses to Questions

**Comments to the Authors:**

Reviewer #1: In this work by Brown and colleagues, the authors examine how distinct types of sleep loss affect sleep homeostasis. They focus on the idea that starvation-induced sleep loss does not result in subsequent sleep rebound because it is accompanied by deeper sleep during the period of starvation. They also begin to examine the mechanisms controlling these processes. The manuscript is interesting but could have gone further with regard to how Dilps connect to other sleep-wake systems known to affect both metabolism and sleep homeostasis. I had some major concerns about the manuscript and data as well.

Overall, the manuscript is not easy to follow. The authors should try to clarify the rationale of each new set of data and work harder to connect them with each other (connecting paragraphs and sections). Also, a majority of the data is in Supplemental figures, much of which seems as important as the main figures. It would help if more of the data is presented as main figures rather than supplemental.

Major points:

1. A major critique is that sleep rebound is measured over 12 hours of the recovery day. Work from many labs makes clear that changes might be washed out over this time frame. Given this importance, measurements of sleep duration, rebound, and depth should be broken down into bins, showing at a minimum whether the absence of change with starvation holds when looking at hours 0-1 of recovery, 0-3, and 0-6. That can all be incorporated to make Fig 1 a more in depth report of the data (rather than only show min/hour as averaged over all 12 hours). This should include sleep traces.

2. Why is sleep loss only measured over the night if 24 hrs of sleep restriction was used? And why was 24 hrs of sleep deprivation chosen rather than 12 hrs of night only, given that day and night sleep are likely different based on many lines of evidence? If day sleep loss with each manipulation is different, this could easily be a driver of differences reported regarding sleep depth at night, and next day recovery. As in point 1, this data could be shown in sleep traces.

3. The LL data (Fig 1D) is flawed regarding rebound. These flies are going to be arrhythmic so will redistribute sleep between night and day, which then will appear to show an increase in sleep over 12 hours of day. However, this is not necessarily rebound sleep, and instead might reflect a change in when sleep occurs from a circadian perspective.

4. The authors show that length of starvation drives changes to sleep depth. How does length of starvation affect sleep rebound duration? After 36 hours of starvation, does sleep rebound now emerge? 48 hrs? This depends on how long they can be starved, of course. My main question (related also to point 1) is whether starved flies ever show sleep rebound.

5. The authors report increased night sleep depth on sucrose compared to other foods (Fig 3C). Does sucrose also result in increased sleep depth across the 24 hour day, or only during night? Would be interesting to know whether the day/night sleep depth (which can depend a lot on background according to published work) is more or less robust depending on the food.

6. What happens to sleep duration during recovery in schematic of Fig 3A? Even though sleep duration is not changed on sucrose (no sleep loss), those flies sleep more deeply. Does this affect subsequent next day sleep in some way?

7. The 2DG experiments omit a critical part in the comparison to starved flies: do they exhibit rebound from a sleep duration perspective? If so, then it calls into question whether this phenocopies starved flies. In general, the 2DG line of logic is not presented clearly. The authors need to add another 1-2 lines of rational to explain how this addresses the contribution of sensory processing to sleep/arousal. I was not able to follow the concluding sentence of this paragraph for this reason. As it stands, this result (Fig S5) seems like a distraction. Moreover, this seems like an indirect way to ask whether the sleep depth changes emerge from sensory differences (tasting certain foods drives different sleep) as opposed to metabolic changes. Can the authors show more directly that sleep changes are related to sensory differences?

8. As with the point above, I do not understand the contribution of Fig S6. Here the authors show that sleep in the SAMM is the same as sleep measured in Fig S5, and that metabolic inhibition with 2DG inhibits metabolism as measured in the SAMM. Is this all just proof of the SAMM system working basically?

9. For Dilp2 RNAi, what about day sleep duration? Is this changed, which could affect sleep depth at night?

10. Also, For Dilp2 RNAi, is it correct that the lack of deeper sleep during recovery (Fig 4G) is unexpected, compared to other forms of sleep loss that induce more, deep sleep (Fig 1C-E)? Can the authors comment on this? Same question for Fig 4N. With caff or LL (Fig S7), they now show increased sleep duration and depth during recovery. Why?

11. Overall, the authors, who were involved with previous work on how different forms of sleep loss affect sleep rebound, missed an opportunity to integrate OA, Dilps, metabolism, and sleep rebound. OA neurons feed into Dilps to affect sleep. Why not examine this in more depth by looking at how OA-based sleep loss (Tdc2>TrpA1) affects recovery arousal, arousal during sleep loss, interactions between different types of food, and metabolic rate, etc? Do Tdc2>TrpA1 flies at elevated temps (at which some sleep persists, ie not total sleep deprivation) show deeper sleep like starved flies, explaining why there is not subsequent rebound? With this manipulation, according to work this group contributed to, flies show persistent memory impairments, as opposed to starved flies. The authors should at least test whether Dilp2 manipulations during starvation (no increased sleep depth) DO show a subsequent memory deficit. And, then the authors can discuss how OA, Dilp2, and other systems are potentially coupled (or dissociable).

Minor:

12. Use of word “remarkably” in Abstract seems like an overstatement

13. In Summary, “surprisingly little is known” also seems like an overstatement given that we now know a decent amount regarding recovery sleep mechanisms.

14. Lines 72-73: there must be a typo. Did the authors mean “…how different genetic…manipulations *that* differentially modulate sleep impact quality and homeostasis.”? or “…how different genetic…manipulations differentially modulate sleep quality and homeostasis.”

15. Fig S1C is referred to in the Results section out of order (after Fig S2), which caused a great deal of confusion while reading. Please move this to Fig S2 or on its own. Also, is the y axis labeled incorrectly? Should this be arousal? Is this different for starvation data than Fig 2E, or replotting the same data?

16. What is “NS” in Fig S2E measuring? Is there no significant difference in arousal in day vs night on any day?

17. Line 145: authors say “nighttime specific increase in sleep depth” immediately after presenting in CS flies that shows increase in sleep depth across day and night. Consider rephrasing to “increase in sleep depth during the night…is due to either…”.

18. Is it correct that sleep duration measures in the SAMM system with 2DG show the same result as Fig S5? If so, the authors should just directly state that they get the same result in both, then move to the metabolic measures.

19. For schematic in 3A, do not show the recovery period since this is not examined.

20. In Fig 4I, what is “w1118>dilp2null”? H, I should match K.

21. Line 229: no rebound in sleep duration or depth after what manipulation? Sucrose feeding? If so, the control data needs to come when sucrose data is first presented. If something different, the authors need to clarify in the text, as this is difficult to follow. Fig legend (line 883) says the manipulation on sucrose “does not change daytime sleep duration”. After moving back to regular food? Hard to know what is meant here.

22. Lines 262-263: cellular basis of the homeostat is poorly understood. This is a surprising statement given many in depth papers on dFB and EB recently, at the molecular and cellular level.

23. Line 282: The SNAP system is different than mechanical shaking using the vortexer, which is what most labs use. Judging from methods sections of fly sleep papers, few use the SNAP system.

24. Line 116 concludes that arousal changes are not the result of "circadian regulation", but the premise of the experiment is to show that changes are not from habituation to the stimulus. The conclusion is really the habituation point, not anything about circadian regulation.

25. Lines 232-234 end with "the resulting homeostatic rebound is independent of sleep depth". This is very confusing - my understanding of the manuscript is the whole point is that whether or not homeostatic rebound occurs depends on prior sleep depth.

26. Line 242: the authors say "no difference in sleep" in dilp null flies, but what they mean I believe is "no difference in night sleep duration", since in fact they do show a difference in sleep depth in these flies compared to controls using other assays.

27. Line 264: nemuri is sleep-promoting, not wake-promoting.

28. Based on Fig 3, how can the authors conclude that loss of protein per se drives increased sleep depth, as opposed to any diet that decreases metabolic rate also increasing depth? The Dilp experiments do not dissociate this because Dilps seem to be required for dietary changes to alter sleep depth. I would particularly consider changing how strongly this is stated in line 92 ("critical role").

Reviewer #2: This is a very interesting and well written paper on the specific role of dilp2 in sleep regulation. It makes a significant contribution to our understanding of the interaction between diet, metabolism and sleep. I just have a few suggestions for minor corrections.

1. Check the references - eg. reference #73 has the wrong author list.

2. Figures - the figures are a little confusing and would benefit from clearer labelling and legends. For example, in figure 4, it is difficult to determine which panels represent day vs night sleep and it is unclear what is the difference between panels D,E and F,G.

3. Genetic background of fly stocks - although the authors state that they use 2 different backgrounds (w1118 and CS) it is not clear if the GAL4/UAS stocks have been backcrossed to the appropriate genetic background. Please clarify.

Reviewer #3: Uploaded as an attachment

**Have all data underlying the figures and results presented in the manuscript been provided?**

Reviewer #1: Yes

Reviewer #2: Yes

Reviewer #3: Yes

PLOS authors have the option to publish the peer review history of their article (what does this mean?). If published, this will include your full peer review and any attached files.

Reviewer #1: No

Reviewer #2: No

Reviewer #3: No

---

## [Decision Letter · Decision Letter 1]

6 Dec 2019

Dear Dr Keene,

We are pleased to inform you that your manuscript entitled "Drosophila insulin-like peptide 2 mediates dietary regulation of sleep intensity" has been accepted in principle for publication in PLOS Genetics, contingent on a small number of minor textual revisions as suggested by Reveiwers #1 and #3 in their final evaluation. Please make these changes as you prepare your final draft for the production team. Congratulations!

Yours sincerely,

Scott D Pletcher

Guest Editor

PLOS Genetics

Gregory P. Copenhaver

Editor-in-Chief

PLOS Genetics

Comments from the reviewers (if applicable):

The authors have adequately addressed nearly all of the reviewer's comments and all agree that this work is appropriate for publication in PLoS Genetics. I recommend acceptance contingent on minor corrections as noted by Reviewers #1 and #3 in their final comments.

Reviewer's Responses to Questions

**Comments to the Authors:**

Reviewer #1: In this revision, the authors have made major changes to address my concerns. This is a really nice manuscript and will have an impact on the field. I have 2 minor points.

1. I think Fig 1G shows pretty clear evidence of some form of sleep duration rebound after starvation during the first 1-2 hours. It stands out by eye, and other readers will notice this once published. I suggest the authors acknowledge this difference/trend somewhere in the paper, because otherwise it will just raise lots of questions after the fact! Maybe add something to the discussion - it doesn't undermine the great work in this paper. There might be a small rebound of sleep duration, even though largely this is attenuated secondary to increased sleep depth during starvation. In my opinion, saying something in the manuscript just makes the body of work stronger overall. But I defer to the authors on this matter.

2. The manuscript still needs a close editorial proofing. I noticed two typos in my reading. Line 152, I believe the word "no" is missing prior to "change", which is a key omission; Line 176 I think should read "inhibition of glycolysis".

Reviewer #2: All issues that I raised in the first review have been adequately addressed.

Reviewer #3: The authors have substantially improved their manuscript and adequately addressed my concerns. I have one minor comment that should be addressed:

1. Per the authors’ response to Reviewer #1’s comments, please make sure all instances where LL experiments/data were previously mentioned have been removed from the manuscript (e.g., see lines 275 and 392-395).

**Have all data underlying the figures and results presented in the manuscript been provided?**

Reviewer #1: Yes

Reviewer #2: None

Reviewer #3: Yes

PLOS authors have the option to publish the peer review history of their article (what does this mean?). If published, this will include your full peer review and any attached files.

Reviewer #1: No

Reviewer #2: No

Reviewer #3: No

**Data Deposition**

http://datadryad.org/submit?journalID=pgenetics&manu=PGENETICS-D-19-01002R1

**Press Queries**

---

## [Editor Report · Acceptance letter]

27 Feb 2020

PGENETICS-D-19-01002R1 

Drosophila insulin-like peptide 2 mediates dietary regulation of sleep intensity 

Dear Dr Keene, 

We are pleased to inform you that your manuscript entitled "Drosophila insulin-like peptide 2 mediates dietary regulation of sleep intensity" has been formally accepted for publication in PLOS Genetics! Your manuscript is now with our production department and you will be notified of the publication date in due course.

With kind regards,

Kaitlin Butler

PLOS Genetics

On behalf of:
